# Widespread misregulation of inter-species hybrid transcriptomes due to sex-specific and sex-chromosome regulatory evolution

Santiago Sánchez-Ramírez[1]*, Jörg G. Weiss[1], Cristel G. Thomas[1,2], Asher D. Cutter[1]*

**1** Department of Ecology and Evolutionary Biology, University of Toronto, Toronto, Canada, **2** Los Alamos National Laboratory, Los Alamos, New Mexico, United States of America

* santiago.snchez@gmail.com (SSR); asher.cutter@utoronto.ca (ADC)

**Data Availability Statement:** Raw sequencing data can be found on the Short Read Archive under accessions SRR12532505- SRR12532522 and BioProject PRJNA659254. Raw gene counts,

## Abstract

When gene regulatory networks diverge between species, their dysfunctional expression in inter-species hybrid individuals can create genetic incompatibilities that generate the developmental defects responsible for intrinsic post-zygotic reproductive isolation. Both *cis*- and *trans*-acting regulatory divergence can be hastened by directional selection through adaptation, sexual selection, and inter-sexual conflict, in addition to cryptic evolution under stabilizing selection. Dysfunctional sex-biased gene expression, in particular, may provide an important source of sexually-dimorphic genetic incompatibilities. Here, we characterize and compare male and female/hermaphrodite transcriptome profiles for sibling nematode species *Caenorhabditis briggsae* and *C. nigoni*, along with allele-specific expression in their $F_1$ hybrids, to deconvolve features of expression divergence and regulatory dysfunction. Despite evidence of widespread stabilizing selection on gene expression, misexpression of sex-biased genes pervades $F_1$ hybrids of both sexes. This finding implicates greater fragility of male genetic networks to produce dysfunctional organismal phenotypes. Spermatogenesis genes are especially prone to high divergence in both expression and coding sequences, consistent with a "faster male" model for Haldane's rule and elevated sterility of hybrid males. Moreover, underdominant expression pervades male-biased genes compared to female-biased and sex-neutral genes and an excess of *cis-trans* compensatory regulatory divergence for X-linked genes underscores a "large-X effect" for hybrid male expression dysfunction. Extensive regulatory divergence in sex determination pathway genes likely contributes to demasculinization of XX hybrids. The evolution of genetic incompatibilities due to regulatory versus coding sequence divergence, however, are expected to arise in an uncorrelated fashion. This study identifies important differences between the sexes in how regulatory networks diverge to contribute to sex-biases in how genetic incompatibilities manifest during the speciation process.

workflow scripts, and main data tables and files can be found on GitHub under the following repository: https://github.com/santiagosnchez/competitive_mapping_workflow.

**Funding:** This work was funded by a Discovery Grant from the Natural Sciences and Engineering Research Council of Canada to ADC and a Postdoctoral Fellowship from the Department of Ecology and Evolutionary Biology at the University of Toronto to SSR. The funders had no role in study design, data collection and analysis, decision to publish, or preparation of the manuscript.

**Competing interests:** The authors have declared that no competing interests exist.

## Author summary

As species diverge, many mutations that affect traits do so by altering gene expression. Such gene regulatory changes also accumulate in the control of static traits, due to compensatory effects of mutation on multiple regulatory elements. Theory predicts many of these changes to cause inter-species hybrids to experience dysfunctional gene expression that leads to reduced fitness, disproportionately affecting genes biased toward expression in one sex and that localize to sex chromosomes. Our analyses of genome-wide gene expression from *Caenorhabditis* nematode roundworms support these predictions. We find widespread rewiring of gene regulation, despite the extensive morphological stasis and conserved expression profiles that are hallmarks of these animals. Misregulation of expression in inter-species hybrids of both sexes is most severe for genes linked to the X-chromosome, but male organismal phenotypes are most disrupted in hybrids. This fragility of male genetic networks and sex differences in regulatory evolution of local versus distant elements may underlie feminized and sterile phenotypes among hybrids. Our work clarifies how distinct components of regulatory networks evolve and contribute to sex differences in the manifestation of genetic incompatibilities in the speciation process.

## Introduction

Many kinds of reproductive barriers can contribute to speciation [1,2], with genetically intrinsic post-zygotic barriers a kind that makes speciation irreversible. Such intrinsic barriers result from disrupted developmental programs due to divergence in the regulatory controls of and functional activity within genetic networks. Consequently, research for decades has aimed to decipher the identity and general features of genetic changes that accumulate by selection and genetic drift to lead to Dobzhansky-Muller (DM) incompatibilities in hybrids of diverging populations, due to non-additive, negatively-epistatic interactions among genes [1,3]. It is therefore crucial to decipher how genes and gene expression evolve to understand how gene regulation influences post-zygotic reproductive isolation through misregulated gene interactions in hybrids [3–5].

Evolution of the regulatory controls over gene expression influences much phenotypic evolution [5,6], despite stabilizing selection as a prevailing force acting to preserve expression profiles [7–11]. Expression differences between species accrue in predictable ways. Regulatory differences between species disproportionately involve the evolutionary accumulation of mutations to *cis*-regulatory elements, facilitated by such changes being predisposed to additivity and having low pleiotropic effects on traits and fitness [12,13]. In contrast, larger, more pleiotropic effects can result from *trans*-regulatory changes that occur at distant genomic positions, such as to transcription factors, chromatin regulators, and small RNA genes. Consequently, theory predicts *trans*-regulatory mutations to fix less readily and to contribute fewer differences between species, despite their large mutational target size and disproportionate contribution to genetic variation within a species [12–15]. Studies nevertheless commonly find both *cis*- and *trans*-regulatory differences between species [16–19]. Indeed, the coevolution of changes to both *cis*- and *trans*-acting factors represents one plausible outcome of stabilizing selection on expression level. The compensatory effects of such coevolved *cis*- and *trans*-regulatory changes yield an overall conserved expression profile [10,14,20,21], but this multiplicity of changes are predisposed to generating misexpression in $F_1$ hybrids due to dysfunctional *cis*-by-*trans* regulatory interactions [5,22].

Decomposing the changes of gene networks into their *cis*- and *trans*–regulatory components, however, presents a challenge to studying gene regulatory evolution. One way to address this problem is with hybrid cross experiments that assess differential expression between two closely related species and from allele-specific expression (ASE) in their $F_1$ hybrids [5,10,23]. Differences in gene expression between species reveal the joint effects of *cis*- and *trans*-regulatory divergence, whereas differences in ASE within $F_1$ hybrids typically represent the effects of *cis*-regulatory divergence alone [23,24]. Studies of this kind have unveiled broad empirical patterns of regulatory evolution, whether carried out in flies [16,25], mice [26], plants [17,18], or yeast [19,27]. Overall, previous work has shown substantial regulatory divergence in both *cis* and *trans*, extensive non-additivity, and disrupted regulation and misexpression in $F_1$ hybrids. Whether these patterns hold for the nematode phylum is, as yet, unknown, and the links between regulatory mechanisms and sex-biases in expression remain incompletely resolved.

Hybrid dysfunction of developmental programs, organismal traits, and fitness may often trace their origins to gene misregulation, from transcriptional to post-translational levels [5,28]. Sex-biased misregulation, therefore, should underlie sex-biased developmental and fitness effects in hybrid individuals. Misexpression of male-biased genes and genes related to spermatogenesis links regulatory disruption to male sterility in hybrids, with supporting empirical evidence in several kinds of animals [26,29–31]. In organisms with chromosomal sex determination, more severe defects typically occur in hybrid individuals carrying heterogametic sex chromosomes (i.e., XO males in *Caenorhabditis* nematodes). This Haldane's rule pattern can arise from dominance effects [32], faster molecular evolution of genes with male-biased expression [33], greater sensitivity of male developmental programs to perturbation (i.e., "fragility") [33], and faster evolution of sex-linked loci [34], among other causes [35,36]. Because of the prominent role that the X-chromosome plays in reproductive isolation [37–39], we might also expect to find greater expression divergence and misexpression for X-linked genes compared to autosomes [5,40–42], with the caveat that genes with male-specific expression might not necessarily be abundant on the X-chromosome [43–45]. Thus, distinguishing between abnormal expression in hybrids for X-linked genes overall and for sex-biased autosomal genes is important to decipher the genetic mechanisms that underpin Haldane's rule in particular and hybrid dysfunction in general.

*Caenorhabditis* nematode roundworms provide an especially tractable system to study speciation genomics [36]. The growing number of *Caenorhabditis* species known to science conform to the biological species concept, with a few cases where sibling species can produce some viable and fertile adult hybrid offspring [46,47]. The *C. briggsae* × *C. nigoni* species pair is one such case, where recent divergence (~3.5 Ma [48]) allows them to form hybrids of both sexes. In this system, *C. nigoni* has the ancestral gonochoristic reproductive mode (XX females and XO males) whereas XX hermaphrodites in the androdioecious *C. briggsae* (also with XO males) are morphologically female and capable of self-fertilization due to each gonad arm producing a set of sperm cells prior to an irreversible switch to oocyte production. In hybrids, both Haldane's rule and Darwin's corollary are fulfilled: $F_1$ male hybrids are always sterile with near complete inviability when arising from *C. briggsae* mothers [49–51]. In contrast, $F_1$ females from both reciprocal crosses are fertile, retaining the "female" condition (i.e., are not hermaphrodites) [49]. This transition in reproductive mode led to the evolution of a suite of genomic and phenotypic traits in *C. briggsae* that define a "selfing syndrome" due to relaxed selection on male function and adaptation to a selfing lifestyle [52,53], and has led to the disproportionate loss of male-biased genes in the *C. briggsae* genome [48,52,54–57]. Moreover, extensive X-linked regions with loci causing hybrid male sterility implicates a large-X effect [58]. Further analyses revealed that X-autosome incompatibilities involve misregulation of the 22G class of small RNAs, leading to down-regulation of spermatogenesis genes and

contributing to hybrid male sterility [31,59]. The full extent of hybrid misexpression throughout the genome of each sex and its root causes in regulatory divergence, however, remain to be characterized.

Given the extensive phenotypic stasis within the genus [60] and previous work showing prevalent action of *cis* and *trans* compensatory evolution, even between more distantly related species [20], we expect to find substantial developmental systems drift within this system, where sex-biased genetic networks will be disproportionately prone to misregulation and misexpression. In particular, we expect greater dysfunctional expression in hybrid males, which are sterile, compared to hybrid females, which are fertile. These effects also are expected to associate most strongly with genetic networks that most depend on X-linked genes. To test these ideas, we analyze mRNA transcriptome expression for each sex from each of *C. briggsae*, *C. nigoni*, and their $F_1$ hybrids. Using ASE profiles, we then characterize and quantify *cis-* and *trans*-acting regulatory causes of expression divergence linking genomic change to sex-biased expression, chromosomal features, and hybrid dysfunction.

## Results

### Extensive expression divergence between species and between the sexes involve the X-chromosome

Each species and sex show distinctive overall transcriptome profiles that are further distinguishable from each sex of $F_1$ inter-species hybrids (**S1A Fig**). *C. briggsae* hermaphrodites resemble females phenotypically except that they are able to produce sperm in addition to eggs, and therefore show masculinized expression of some genes (**S1A, S2** and **S3 Figs**). To more appropriately contrast "female" expression profiles between hermaphrodites and females, we identified 1,238 orthologous genes that are male-biased and up-regulated in hermaphrodites relative to sperm-less *C. briggsae* "pseudo-female" mutants (*she-1(v4)*; [54]) (9% of the 13,636 one-to-one orthologs analyzed). These 1,238 genes were then analyzed separately from most sex-based analyses that, for simplicity, we refer to hereafter as "female" expression contrasts.

The transcriptomes of pure *C. briggsae* and *C. nigoni* differed in expression for more than half of their genes, with slightly more genes differentially expressed for "females" than for males. Females had a total of 61% (7,598) of genes differentially expressed between species, compared to 54% (7,385) for males (**Fig 1A**). The X-chromosome shows the most extreme differences in number of differentially expressed genes between species, with both males and "females" showing significantly higher ratios of upregulated X-linked genes in *C. nigoni* than in *C. briggsae* (**Fig 1B**). Autosomes, by contrast, showed greater abundance of genes with higher expression in *C. briggsae*, albeit only significantly for Chromosome I in males (**Fig 1B**; Fisher's exact test; $P < 0.05$). Similarly, we found more genes with significant ASE for autosomes among female hybrids (6,070 or 61%; n = 9,932) than among hybrid males (5,402 or 53%; n = 10,155). The enrichment of genes where a given allele gets upregulated in hybrids was largely consistent across autosomes of hybrid males and females (**Fig 1B**), albeit the pattern was slightly more extreme for females (higher expression of *C. briggsae* allele: 3,215 or 32%; *C. nigoni* allele: 2,855 or 29%) than for males (higher expression of *C. briggsae* allele: 2,793 or 27%; *C. nigoni* allele: 2,609 or 26%). The slightly greater tendency in females for *C. briggsae* alleles to be upregulated could suggest more regulatory changes fixed in the *C. briggsae* lineage leading to higher expression, or some degree of mapping bias in *C. briggsae*-derived reads compared to those from *C. nigoni* (simulations suggest that mapping biases are an unlikely cause; see Material and Methods).

Within each species, approximately 60% of genes showed significant sex biases in expression (**Fig 1C**). Male-biased and female-biased genes occurred with similar incidence in both

species: 30% *C. nigoni* male-biased, 29% *C. briggsae* male-biased, 29% *C. nigoni* female-biased, 27% *C. briggsae* "female"-biased. In stark contrast, over 80% of genes in F$_1$ hybrids were significantly sex-biased, with a slightly higher incidence of male-biased genes (43% vs 40%). In *C. briggsae* and F$_1$ hybrid transcriptomes, Chromosomes I and III were enriched for "female"-biased genes, whereas Chromosomes V and X were enriched for male-biased and sex-neutral genes (**Figs 1D and S2**). None of the *C. nigoni* chromosomes exhibited strong enrichment of sex-biased genes and genes with "female"-biased expression in both species and in hybrids showed strong depletions from the X-chromosome (**Figs 1D and S2**).

### Expression dominance in F$_1$ hybrids differs between males and females

We contrasted expression profiles of F$_1$ hybrids with their parent species to infer the expression inheritance of genes, i.e., to identify genes that exhibited additive, dominant (*C. briggsae*- and *C. nigoni*-like expression), or transgressive (overdominant and underdominant) expression patterns for each sex (**Fig 2A**). Gene sets with distinct expression inheritance profiles

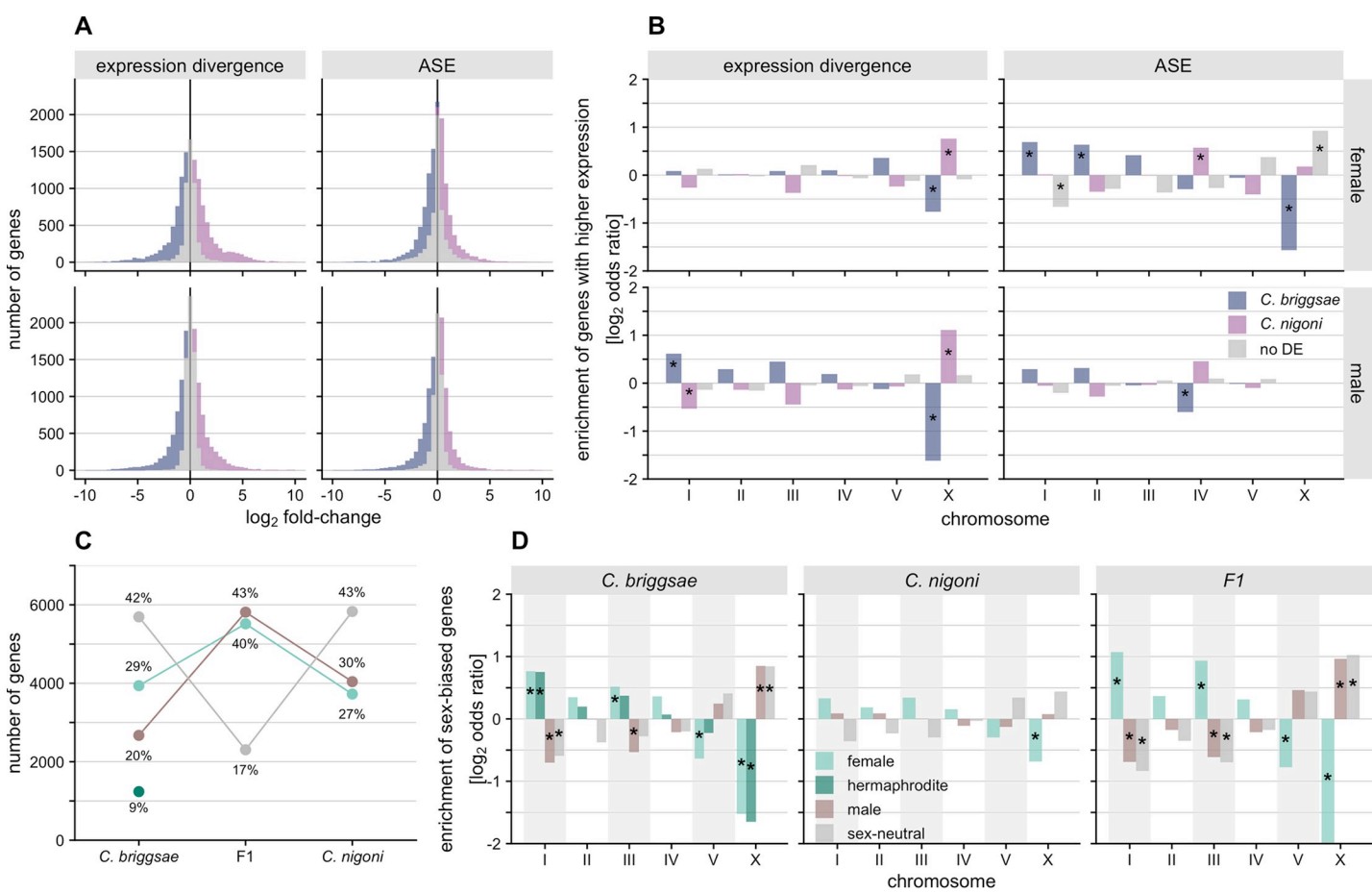

**Fig 1. Incidences of differentially expressed genes for 13,636 orthologs between species and sexes.** (**A**) Histogram of log$_2$ expression divergence between species and allele-specific expression (ASE) in hybrids (*C. nigoni*/*C. briggsae*) for female (top row) and male transcriptomes (bottom row). Colored bars indicate significantly differentially expressed genes, while grey bars indicate counts of genes with non-significant expression differences. (**B**) Enrichment of differentially expressed genes between species and alleles for females and males (log$_2$ odds ratio, i.e. observed/expected). Asterisks mark significant enrichments (positive values) or depletions (negative values) on chromosomes (Fisher's exact test, $P$ value < 0.05 and |log$_2$ odds ratio| > 0.5). On the legend, "no DE" denotes genes that are not significantly differentially expressed. (**C**) Number and percentage of genes that show significant sex-bias is greater in F$_1$ hybrids than parent species. In *C. briggsae*, male-biased genes also expressed in hermaphrodites (dark green) are shown separately from male-biased only (brown) and from female-biased genes expressed in hermaphrodites (light green, "female") (see Fig 6). (**D**) Enrichment of genes with significant sex-bias and sex-neutrality in parent species and F$_1$ hybrids for each chromosome.

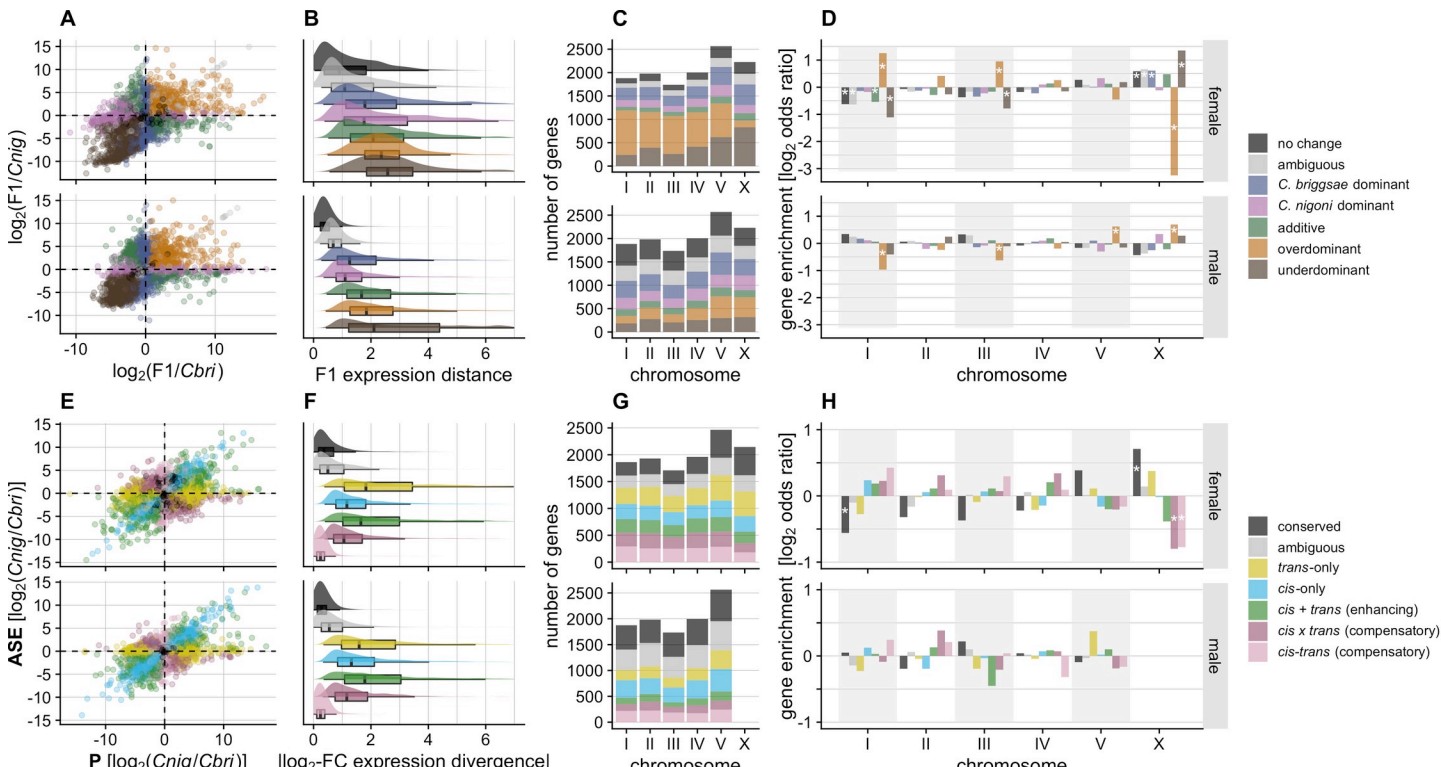

**Fig 2. Sex-specific differences in expression divergence and regulatory controls identify expression inheritance profiles between species.** (**A**) Per-gene biplot of log₂ expression differences between F₁s and each parent species. (**B**) Box- and density plots of expression distance from the origin or centroid of F₁ hybrids for genes within a given expression inheritance category (see Material and Methods). (**C**) Stacked barplot of gene counts in each expression inheritance group for each chromosome. (**D**) Per-chromosome enrichment (log₂ odds ratio, i.e. observed/expected) of genes in a given expression inheritance group. Asterisks mark significant enrichments (positive values) or depletions (negative values) on chromosomes (Fisher's exact test, *P* value < 0.05 and |log₂ odds ratio| > 0.5). (**E**) Biplot of expression divergence between species (x-axis) versus allele-specific expression (ASE) in hybrids that indicates the magnitude of *cis*-acting expression difference between alleles (y-axis). (**F**) Box- and density plots of the magnitude of absolute expression divergence between species for each type of *cis* and *trans* regulatory changes. (**G**) and (**H**) as for C and D, but indicating different types of *cis* and *trans* regulatory-change profiles. Colors indicate different groups of genes with different expression inheritance (see legend for A-D) and *cis* and *trans* regulatory changes (see legend for E-H). Top panels in each of A-H correspond to female transcriptomes, bottom panels to male transcriptomes.

revealed substantial differences between the sexes in terms of expression distance (**Fig 2B**), number of genes (**Fig 2C**), and enrichment across the genome (**Fig 2D**).

The sexes differed most strikingly in their total number of transgressive genes: 26% (3,494) of genes show transgressive profiles in males (1,788 overdominant and 1,706 underdominant) versus 55% (6,881) in females (4,121 overdominant and 2,760 underdominant) (**Fig 2A and 2C**). Transgressive genes are thought to be associated with hybrid dysfunction, as such underdominant and overdominant expression profiles represent misexpression phenotypes that manifest values beyond the range of both parents [30,61]. Given the pronounced sterility of hybrid males, we were surprised that hybrid male transcriptomes showed only half the incidence of misexpression as hybrid females. This finding suggests that the genetic networks that control fitness are more robust to expression perturbation in female hybrids than in males.

In addition, genes classified as underdominant, overdominant, and additive had significantly higher Euclidean expression distances from the centroid of expression space than genes with no change in expression or with simple dominance (Gamma-distributed multiple generalized linear least square regression [GLM], *P* < 0.001) (**Fig 2A and 2B**). This observation suggests that genes with these expression inheritance profiles are more prone to deviant expression phenotypes. However, except for transgressively expressed genes, "females" had on

average significantly larger magnitude in expression distance from the centroid expression space than males (GLM, all $P < 0.001$ except transgressive $P > 0.1$) (**Fig 2A and 2B**). These results are consistent with our multidimensional scaling analysis that showed shorter expression distances for $F_1$ males to parental males (**S1A Fig**), in contrast to more dissimilar expression profiles of F1 females to parental females.

Genes with additive expression of alleles from both parental species were relatively rare in $F_1$ hybrids of both sexes (8% or 691 genes in females; 8% or 946 genes in males) (**Fig 2C**). Genes showing simple dominance were up to four-times more common, with approximately 20–30% of genes expressed by each sex either matching *C. briggsae* or *C. nigoni* expression (23% or 2,869 genes in females; 30% or 3,698 genes in males). Expression dominance matching *C. briggsae* was consistently more frequent in hybrids of both sexes (15% or 1,876 genes in females and 18% or 2,228 genes in males), however, compared to expression dominance matching *C. nigoni* (**Fig 2C**; 8% or 993 genes in females and 12% or 1,470 in males), being more extreme in female hybrids across autosomes (mean ratio *C. briggsae* vs *C. nigoni* dominant expression = 1.75 vs 1.14 in males) (**Fig 2C and 2D**). The disproportionate dominance of the *C. briggsae* copy in hybrid females was even more pronounced for the X-Chromosome ($F_1$ female X ratio = 2.66 vs $F_1$ male X ratio = 1.13).

## Expression dominance on the X-chromosome is distinct in hybrids of both sexes

Genes and traits with dysfunctional expression are often associated with the X-chromosome, and we expect differences between the sexes due to Haldane's rule [26,29,40,41]. Consistent with these expectations, we found inheritance profiles associated with misexpression to differ between $F_1$ males and females across the genome, and to differ most conspicuously for the X-chromosome. In particular, the X-chromosome was enriched for underdominant genes in both $F_1$ males and females (**Fig 2C and 2D**), but only significantly so in females (Fisher's exact test, $P < 0.05$). The X-chromosome was also significantly enriched for overdominant genes in $F_1$ males, whereas females had significant depletions of such genes on the X (autosomal enrichments: V in males, and I and III in females) (**Fig 2C and 2D**; Fisher's exact test, $P < 0.05$). These data show clear differences in expression inheritance across chromosomes between the sexes and reflect distinct hybrid expression dynamics between autosomal and X-linked genes.

Given the distinct misexpression of genes linked to the X-chromosome, we evaluated whether disrupted dosage compensation in hybrids might have systematically perturbed X-linked expression profiles. X-chromosome dosage compensation in *Caenorhabditis* acts to halve expression of the two copies in females and hermaphrodites [62]. The magnitude of expression that we observed for the X-linked genes in hybrid females, however, does not differ on average from either parental species (**S1B Fig**). Consistent with the implications of conserved mechanistic features of the dosage compensation complex and its regulation across multiple *Caenorhabditis* species, including *C. briggsae* and *C. nigoni* [63], this suggests that dysfunctional dosage compensation is not a key driver of X-linked underdominant expression in hybrid females.

## *Cis* and *trans* regulatory divergence modulates differences between sexes

Identifying the spectrum of changes to *cis*- and *trans*-acting regulators is important to understand how selection influences the evolution of gene expression and its effects on hybrid phenotypes. Correspondingly, we classified the types of regulatory changes and examined how they perturbed gene expression in $F_1$ hybrids for each sex. Consistent with ASE studies in flies [16,25], mice [26], plants [17,18] and yeast [19,27], we found substantial expression divergence

due to *cis-only* or *trans*-only regulatory changes in addition to joint effects of *cis* and *trans* changes, with changes in *trans* and *cis + trans* conferring larger magnitudes of expression divergence (**Fig 2E and 2F**). Genes with *cis-only* and *trans*-only effects were not significantly enriched on any autosome for either sex, although Chromosome V exhibited a trend toward enrichment of genes with *trans*-only effects for males (**Fig 2H**). Comparing regulatory divergence between sexes, we found nearly 60% more genes involving *trans*-only changes in females (18% or 2,195 genes) compared to males (12% or 1,391 genes) (**Fig 2G**). For genes with *cis*-only divergence, however, the sexes showed a reciprocal pattern: *cis*-only divergence was more prevalent in male than in female transcriptomes (17% or 1,944 genes in males; 14% or 1,647 genes in females). Thus, the expectation that *cis*-regulatory divergence will be more prevalent than *trans*-regulatory divergence holds true only when expression is measured in males, and points to fundamental differences between the sexes in trends of regulatory evolution.

Despite the limitations posed by the hemizygous condition of the X-chromosome in males, we devised a strategy to confidently assign one of three categories of regulatory divergence to X-linked genes in males based on their expression inheritance. In short, we inferred genes to have i) *cis*-only* divergence (inferred from expression dominance with respect to *C. nigoni*, which may include genes with X-linked *trans* regulators, with this distinction denoted by *), ii) *trans*-only* divergence with recessive *C. nigoni trans*-acting factors (expression dominance with respect to *C. briggsae*, which may exclude X-linked genes with codominant *trans* regulators, denoted by *), which provides a lower-bound estimate of *trans*-acting regulatory divergence affecting the X, and iii) *cis-trans* compensatory changes (no difference in expression between parent species with hybrids showing over- or underdominance). We did not find enrichments of *cis*-only* divergent genes on the X, as might be expected under a "large-X" effect model for regulatory contributions to hybrid dysfunction. Similarly, we found only a non-significant trend in both sexes toward enrichment of *trans*-only regulatory divergence affecting X-linked genes (**Fig 3**; Fisher exact test, $P > 0.05$). Instead, however, *cis-trans* compensatory changes were 1.9-fold enriched on the X chromosome for males and 1.7-fold under-represented on the X for females (**Fig 3**; Fisher exact test, $P < 0.05$). These results for regulatory divergence "regroupings" are consistent with the full set of categories that could be defined for females (**Fig 2H**). This finding implicates stabilizing selection on X-linked male-specific networks as being especially prone to disruption in $F_1$ hybrids, and contrasts with the 1.6-fold enrichment of conserved regulatory controls for X-linked genes in females (**Fig 3**; Fisher exact test, $P < 0.05$).

Because of codominant effects of *trans*-acting factors controlling the expression of both alleles, *trans*-only regulatory divergence is often associated with deviations from additivity [64]. Consistent with this idea, we found *trans*-regulatory changes more often associated with genes having dominant expression patterns and that genes with additive expression more often associated with genes having significant *cis*-effects (*cis*-only and *cis + trans*) (**Fig 4**). However, genes with *cis*-only effects often showed dominant patterns of expression. Genes that show *cis*-only divergence on autosomes for male expression often have *C. briggsae* or *C. nigoni* dominant expression in hybrids whereas, in females, they are more typically overdominantly expressed (**Fig 4**). We can think of four potential mechanisms that might contribute to this sex difference and to the *cis*-dominant effect in general: (1) more extensive post-transcriptional regulation in females [65] might limit transcript abundance through epistatic interactions; (2) greater degradation or turnover of *cis*-acting binding sites for male-expressed genes [55,56] could reduce affinity in transcription factor binding activity; (3) the presence of species-specific *trans* regulators that originate from lineage-specific gene duplication and/or loss resulting in allele-specific *trans*-regulation; and (4) some form of transvection that causes

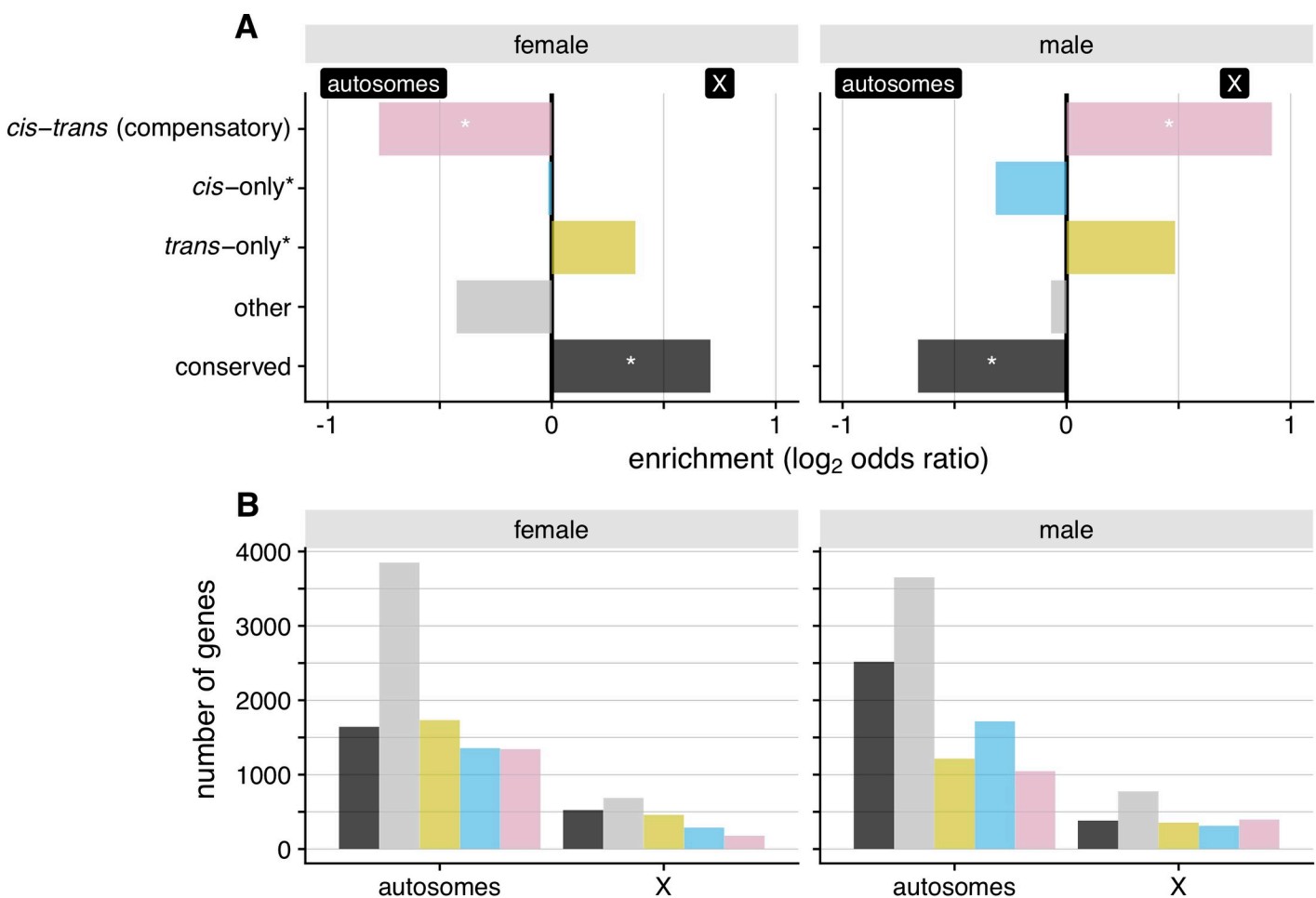

**Fig 3. The X-chromosome is significantly enriched for compensatory *cis-trans* changes in males.** Nominal, but non-significant, enrichments of *trans*-only* regulatory changes are also observed in both sexes. (**A**) Positive values of the log$_2$ odds ratios for females and males denote enrichment for the X-chromosome and negative values indicate enrichment for autosomes. (**B**) Gene counts for autosomes and the X-chromosome for regulatory divergence categories inferred from expression in males and females. In males, X-linked regulatory divergence was classified following S13 Fig (also see Material and Methods). Briefly, X-linked compensatory *cis-trans* changes in males represent genes that are not differentially expressed between species, but hybrids show over- or underdominance; *cis*-only* represent genes with expression divergence between species and hybrids matching *C. nigoni* expression; *trans*-only* denote genes with expression divergence with hybrids matching *C. briggsae* expression; and the "conserved" category refers to genes that are not differentially expressed between species and between species and F$_1$ males. The * differentiates these categories from the *cis*-only and *trans*-only categories inferred from ASE in autosomes and XX individuals.

single-allele expression in hybrids and one of the parent species; this has been identified as a source for the expression of a dimorphic trait in *Drosophila* [66].

Furthermore, we categorized genes into 13 groups based on distinct combinations of species differences and sex differences in expression, including their interactions, and looked at the proportion of genes with different *cis*- and *trans*-effects (**Fig 5A and 5B**). Our results are consistent with the idea that *cis* and *trans* changes each play distinct roles in sex-biased expression and sexual dimorphism [67]. We find, on one hand, that *trans*-only changes in females are more often associated with genes that have male-biased expression (37% *trans*-only in male-biased genes vs. 15% in female-biased genes) and, on the other hand, that female-biased genes show more conserved regulation (32% conserved in female-biased genes vs. 9% in male-biased genes) when expressed in males (**Figs 5D and S2**). This pattern suggests that distinct sex-specific regulatory controls may be asymmetric in this system and may have evolved to repress expression of genes biased for the opposite sex. Together, these observations highlight

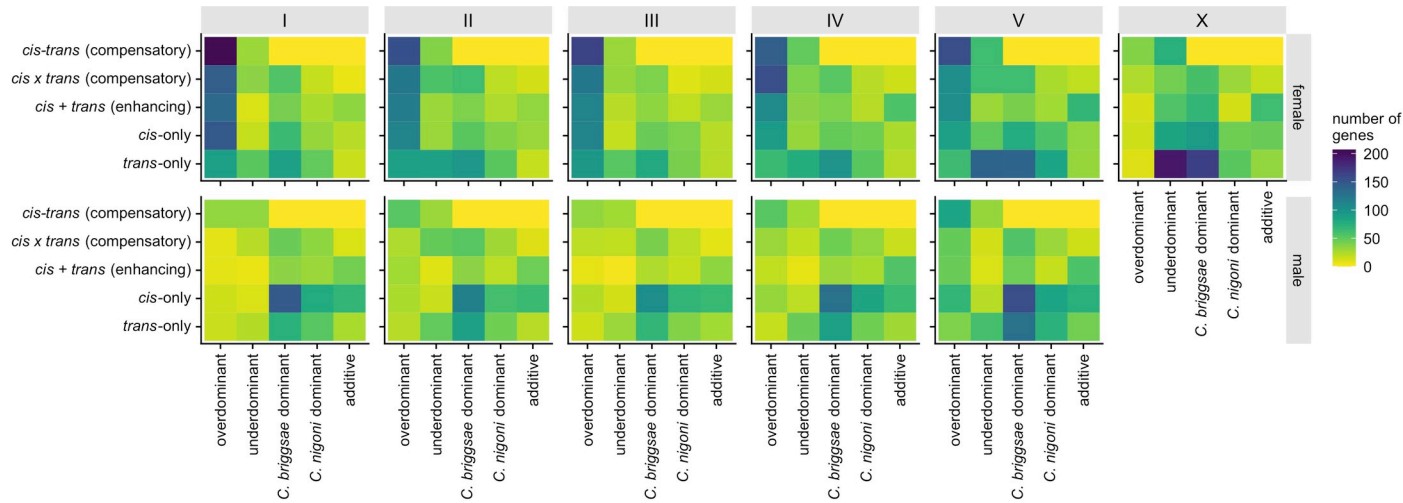

**Fig 4. Differences in regulatory controls between chromosomes and between sexes.** Heatmap of the number of genes in each expression inheritance group (x-axis) for each type of *cis* and *trans* regulatory changes (y-axis) for each chromosome (I-V, X) and each sex. Excludes conserved or ambiguous expression and regulation categories (S3 Fig).

how the evolution sex-specific regulatory controls can underlie sexually dimorphic expression phenotypes, with implications for sex biases in the speciation process.

## Hybrid misexpression commonly involves genes with joint *cis-trans* regulatory changes

Genes exhibiting transgressive expression profiles in $F_1$ hybrids are often associated with dysfunctional traits, due to radically different expression from those of parent species. *cis* and *trans* changes with opposing effects can interact epistatically in hybrids to induce transgressive expression and allelic imbalance [22]. Such regulatory evolution can arise through coevolutionary fine-tuning even when overall expression level is subject to stabilizing selection [6,11]. Consistent with this idea, we found that genes with transgressive effects in hybrids often are associated with *cis-trans* regulatory changes (33% or 877 genes in males and 42% or 2,872 genes in females; **Fig 4**). In both sexes, $F_1$ hybrids revealed a higher fraction of compensatory changes (27% in males and 31% in females) compared to enhancing *cis-trans* changes (6% in males and 11% in females). These results highlight how stabilizing selection can act independently on each sex to maintain sex-specific regulation, leading to opposing *cis* and *trans* effects that induce dysfunctional expression in $F_1$ hybrids. In addition, we also found substantial *cis-trans* compensatory changes among genes that show no change in expression between species (**S3 Fig**) and in those that do not show sex-biased expression (i.e., C-N-N group in **Fig 5D**), implicating extensive developmental systems drift in regulatory processes despite conserved profiles of expression.

Genes with additive expression inheritance also may generate hybrid dysfunction by generating intermediate expression profiles in $F_1$ hybrids and are thought to commonly reflect *cis*-only regulatory divergence [23,40,68]. In line with this idea, we found that genes classified as additive associated more often with significant *cis*-acting divergence in both sexes (*cis*-only: 43% in males, 26% in females; and enhancing *cis-trans* effects: 30% in males, 42% in females; **Fig 4**). However, expression additivity is not abundant in our analysis (**Fig 2C**), suggesting that it is not a major source of phenotypic dysfunction in hybrids of this system.

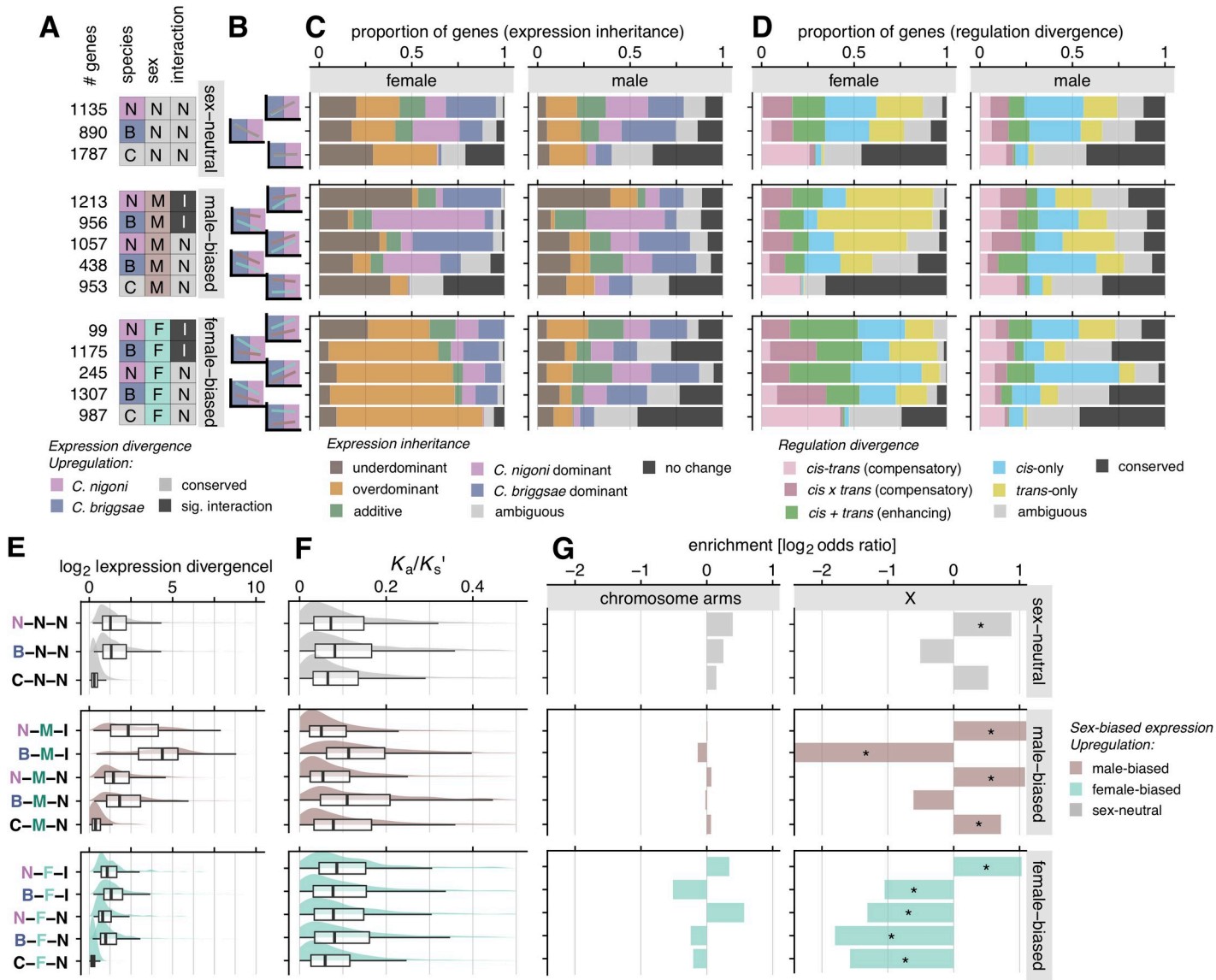

**Fig 5. Male-biased genes show higher expression divergence, molecular evolution, sex-specific regulatory divergence, and X enrichments.** (**A**) Thirteen distinct species-by-sex gene expression gene groups with respective gene counts. Matrix color indicates state with higher relative expression (N = *C. nigoni*, B = *C. briggsae*; M = male, F = female; I = significant interaction; gray cells indicate no difference between treatment groups: C = conserved, N = non-significant interaction). (**B**) Centroid relative expression reaction norm plots for each group indicating expression differences between species and sexes (S4 Fig). (**C**) Proportion of genes within each species-by-sex gene group differ in the relative representation of expression inheritance categories and (**D**) types of *cis*- and *trans*-regulatory divergence, distinctly for females (left panels) and males (right panels). (**E**) Absolute expression divergence and (**F**) protein sequence divergence ($K_a/K_s$) differs across species-by-sex expression profiles most strongly for male-biased genes. (**G**) Male-biased genes are not enriched in chromosome arms but are more abundant on the X-chromosome relative to autosomes (except for the BMI group that is rare on X, see Fig 6). * Fisher's exact test, $P < 0.05$ and |$\log_2$ odds ratio > 0.5|.

To further assess the role that different regulatory controls play in the origin and maintenance of divergent sex-biased expression, we contrasted expression inheritance and patterns of *cis*- and *trans*-regulatory divergence for male-biased and female-biased genes (**Fig 5C and 5D**). We found that male-biased genes expressed in F₁ hybrids of both sexes frequently show underdominant transgressive misexpression compared to sex-neutral and female-biased genes (**Fig 5A–5C**). Examining male-biased genes when expressed in females, we find that they often show expression dominance matching the species with lower expression (**Fig 5A–5C**; N-M-I,

B-M-I, N-M-N, B-M-N; also, M5, M8, and M14 in **S2 Fig**). Additionally, genes that show conserved expression between species, but with significant sex-bias (C-M-N for male-biased and C-F-N for female-biased genes), also often are either misexpressed (over- or underdominant) or have had no significant change in expression in hybrids (**Fig 5C**). These groups also have the highest proportion of genes with *cis-trans* compensatory changes (**Fig 5D**), suggesting that many "conserved" sex-biased networks have undergone substantial developmental systems drift.

Male gene networks may experience greater "fragility" if they are more prone to perturbation by dysfunctional gene interactions. Fragility could arise by higher rates of molecular evolution among male-biased genes or by higher downstream effects of male-specific regulators. We found no overall significant difference between male-biased and female-biased genes for sequence differences in upstream regions (GLM, $1-P_{cons}$ male-biased vs. female-biased, $t =$ -0.48, $P = 0.62$; arms vs centers $t = -27.66$, $P < 0.0001$). We therefore conclude that the effects of upstream regulatory changes exert disproportionately strong effects on male-biased genes, implying that male genetic networks are more fragile. Complementing this idea, genes expressed in $F_1$ males are more commonly underdominant when they correspond to male-biased genes than to female-biased genes (957 genes among male-biased genes vs 431 genes among female-biased genes). Moreover, male-biased genes have a higher proportion of genes with enhancing or compensatory *cis-trans* changes (1,086 or 23% genes in male-biased genes vs 702 or 18% genes in female-biased genes). Male-biased genes also define more distinct expression profile modules than do female-biased genes (6 versus 3 co-expression modules), including one with male-specific underdominance (M15 in **S2 Fig**). By contrast, female-biased genes expressed in $F_1$ females were predominantly overdominant and are more often associated with *cis*-only and enhancing *cis-trans* changes (**Figs 5D, 5C and S2**). These observations suggest that female gene regulatory networks can be more resilient to regulatory divergence, with male networks being more fragile, potentially translating into similar resilience and fragility of organismal traits such as fertility [49].

## Faster regulatory and molecular evolution of male-biased and spermatogenesis genes

Sexual selection and sexual conflict are predicted to drive faster rates of molecular evolution and expression divergence [38,69,70]. Consistent with these predictions, we found that male-biased genes have higher average expression divergence (|log$_2$-fold-change|, GLM, $t = -23.639$, $P < 0.001$) and faster rates of molecular evolution in the two groups of male-biased genes that are rare on the X-chromosome ($K_a/K_s$ for B-M-I: GLM, $t = -8.947$, $P < 0.001$; B-M-N: GLM, $t = -8.764$, $P < 0.001$) (**Fig 5E and 5F**). Male-biased genes show elevated expression divergence compared to sex-neutral genes and female-biased genes as a whole, though the signal for faster sequence evolution was weak ($K_a/K_s$ male-biased vs. sex-neutral GLM, $t = -0.77$, $P = 0.44$; male-biased vs. female-biased GLM, $t = -0.78$, $P = 0.43$; $1-P_{cons}$ male-biased vs. female-biased, $t = -0.48$, $P = 0.62$; arms vs centers $t = -27.66$, $P < 0.0001$). Rapid sequence evolution for male-biased genes was not associated with enrichment in chromosomal arms (**Fig 5G**; Fisher's exact test, $P > 0.05$), regions which are known to show higher divergence [48]. The groups of male-biased genes with enrichments on the X-chromosome, however, have either conserved expression between *C. briggsae* and *C. nigoni* or have higher expression in *C. nigoni* males (**Fig 5G**; C-M-N, N-M-I, N-M-N; **S2 Fig**; Fisher's exact test, $P < 0.05$), indicating that the X-chromosome is home to a subset of genes that reflect ancestral male-biased gene networks.

We observed the highest expression divergence as well as high rates of molecular evolution in the distinctive set of genes that combine male-biased expression, higher expression in *C. briggsae* than *C. nigoni*, and a species-by-sex interaction (B-M-I; **Fig 5E and 5F**). The species-

by sex interaction in this B-M-I group indicates a masculinized expression profile for *C. briggsae* hermaphrodites, implicating a role for them in sperm production (**S4 Fig** and M5 in **S2 Fig**). To test this idea, we looked at *C. elegans* genes previously identified as spermatogenesis genes [45] and found overlapping orthologs in *C. briggsae* and *C. nigoni* to be 11-fold enriched in the B-M-I group (**Fig 6B and 6C**; Fisher's exact test, $P < 0.05$) and depleted from the X-chromosome (**Fig 5G**), consistent with previous observations for sperm genes in *Caenorhabditis* [36,43,44]. Further consistent with sperm-related function, male-biased genes that show upregulated expression in *C. briggsae* hermaphrodites relative to *C. briggsae* pseudo-females (**Fig 6A–6C**) were highly enriched (46-fold) and overlapped extensively with the B-M-I group (**Fig 6B and 6C**; Fisher's exact test, $P < 0.05$). Thus, the high divergence in both expression and sequence for these genes suggests distinctive selection pressure on them, potentially reflecting the outcome of sexual selection and sexual conflict on sperm.

The collection of male-biased genes that also show elevated expression in *C. briggsae* hermaphrodites, putatively linked to spermatogenesis, showed expression phenotypes in hybrids often resembling *C. nigoni*: extensive expression dominance for the *C. nigoni* copy and under-dominant expression in $F_1$ hybrids as well as high expression divergence (**Figs 6D and 6E and S5**). They also show an abundance of *trans*-only regulatory divergence (**Fig 6F and 6G**).

### Regulatory and expression divergence within the sex determination cascade

Sex determination of somatic and germline development involves a negative regulatory cascade [71], including the secreted protein HER-1 that inhibits transcription factor TRA-2, and further downstream, TRA-1 represses genes such as *fog-3* in controlling spermatogenesis [72,73]. Consistent with what is known about this pathway [71], our data shows male-biased expression of *her-1* in both parental species and that it has evolved enhancing *cis* + *trans* regulatory changes that implicate lineage-specific regulatory changes that promote its expression in males of both species (**S6 Fig**). Without HER-1, TRA-3 then cleaves the intracellular domain of TRA-2 in XX individuals, which then interacts with FEM proteins preventing FEM–TRA-1 interactions [73]. We found the ortholog of *tra-3* to be expressed and regulated (*cis*-only) similarly between sexes of pure species and hybrids. However, *fem* genes and *tra-2* seem to be female-biased and overdominantly expressed (**S6 Fig**), suggesting sex-specific co-regulation; and at least two have potentially evolved *cis* and *trans* regulatory changes (*fem-2*, *fem-3*). In addition, *cis* x *trans* regulatory effects on *tra-1* seem to have evolved in females, suggesting that opposing regulatory changes have evolved on an important transcription factor controlling germline and somatic sexual differentiation. However, the story for *tra-1* regulatory divergence in males differs from females and is also less clear: our analysis categorizes it as "ambiguous" (significant ASE, but non-significant regulatory divergence and non-significant *trans* effects). In contrast, orthologs of *fog-3*, which is involved in spermatogenesis, get upregulated in *C. briggsae* hermaphrodites and in males of both species (B-M-I), as expected. In $F_1$ males, however, *fog-3* is misexpressed (under-dominant) having a non-significant *trans* effect ($P = 0.28$) and a marginally-significant *cis* ASE effect ($P = 0.04$). Its expression in $F_1$ females shows *C. nigoni* dominant expression that is due to *trans*-only regulatory divergence between the species (**S6 Fig**). The regulatory divergence between *C. briggsae* and *C. nigoni* for genes involved with sex differentiation and development point to plausible mechanisms for suppression of spermatogenesis in XX hybrids to produce a "female" rather than a "hermaphrodite" phenotype, as well as yielding hybrid male sterility.

### Genome architecture only modestly affects regulatory divergence

Given that protein-coding sequence evolution and gene composition vary non-randomly along chromosomes in many *Caenorhabditis* species in association with the chromosomal

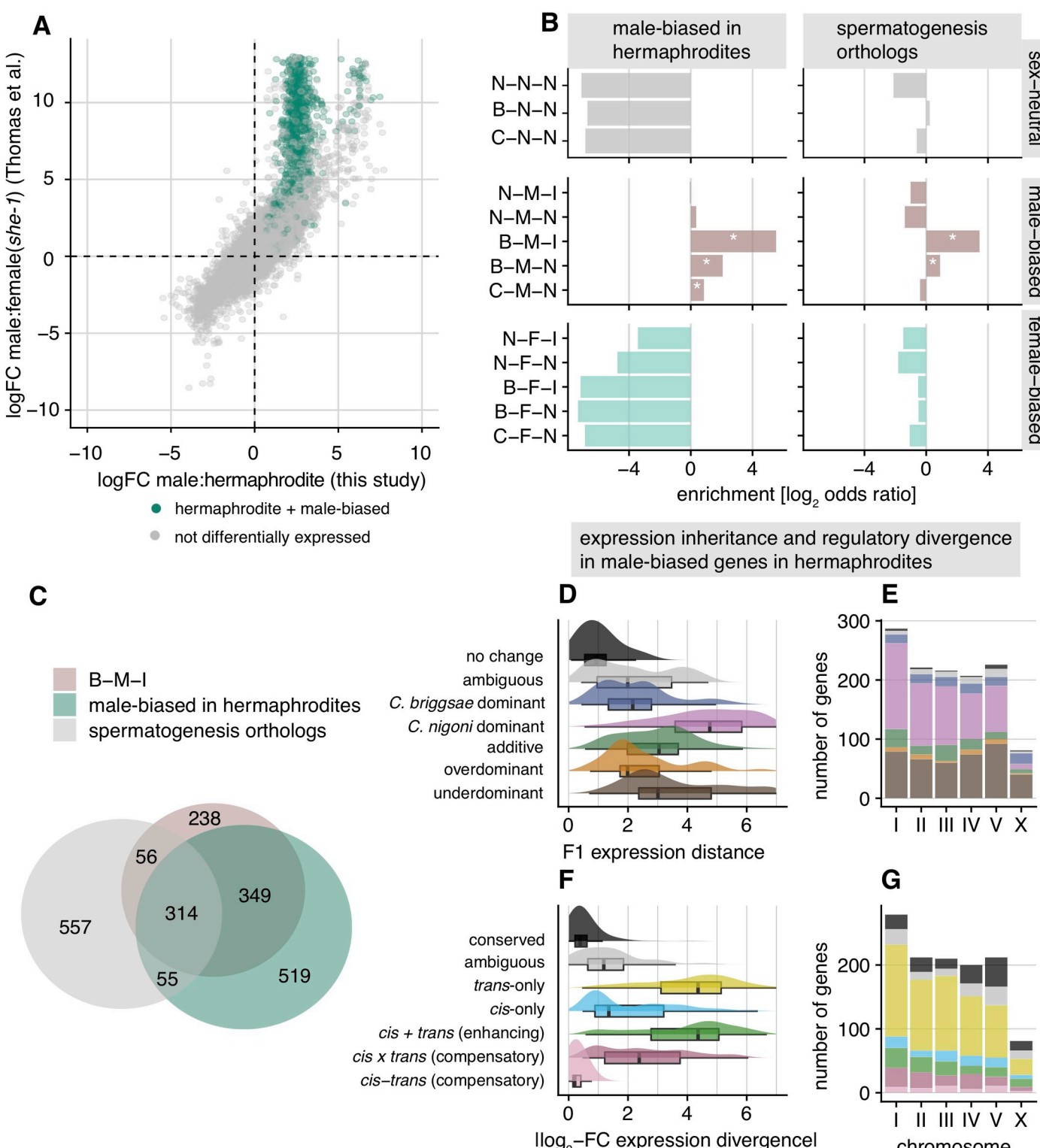

**Fig 6. Male-biased genes that are upregulated in hermaphrodites are enriched for spermatogenesis genes and have distinct expression and regulatory patterns in hybrids.** (**A**) Biplot of log-fold-change values in male:female contrasts (sex-biased expression) between our *C. briggsae* data and data from (54), which includes *C. briggsae* "pseudo-females". Green dots denote genes that were differentially expressed between hermaphrodites and "pseudo-females". (**B and C**) Both male-biased genes upregulated in hermaphrodites and orthologs to *C. elegans* spermatogenesis genes (45) are enriched for the B-M-I group that shows a species-by-sex interaction (Figs 5 and S4). Divergence in expression and regulation (**D and E**) is driven largely by *trans*-only regulatory changes (**F**) resulting in abundant *C. nigoni* dominant expression and underdominant misexpression (**G**) in hybrid females.

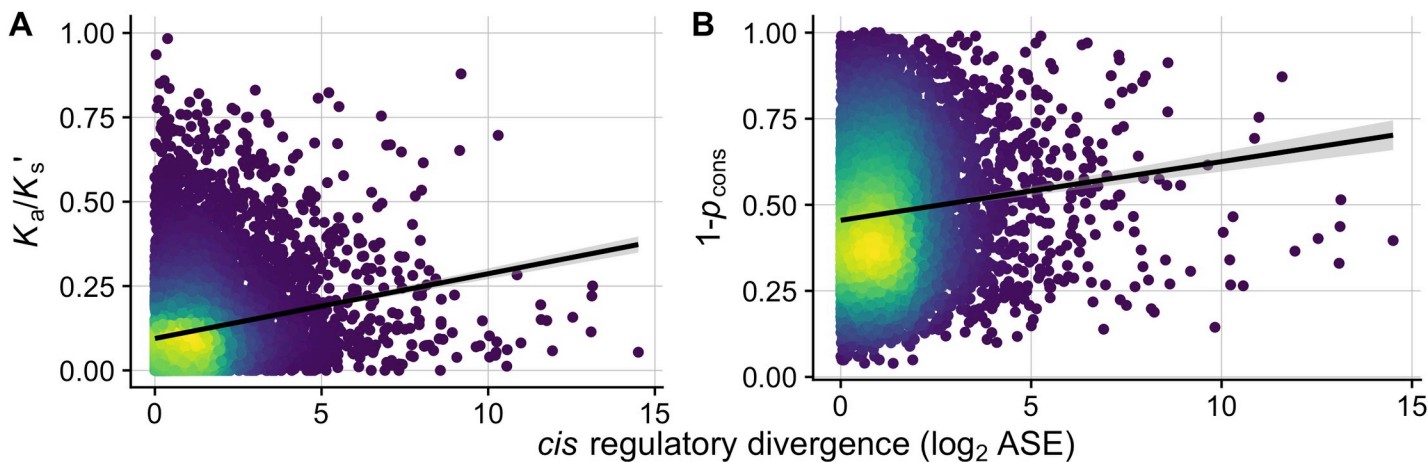

**Fig 7. The magnitude of *cis*-regulatory divergence shows positive but weak correlations with rates of molecular evolution.** (**A**) Scatter plot and linear regression of mutation-adjusted protein evolution rates ($K_a/K_s$', Spearman's $\rho$ = 0.17, $P < 0.0001$; adjusted $R$ = 0.035, $m$ = 0.019, $P < 0.0001$) and (**B**) the proportion of non-conserved 5 bp windows within 500 bp upstream of each gene (1-$P_{cons}$, Spearman's $\rho$ = 0.12, $P > 0.05$; 1-$P_{cons}$: adjusted $R$ = 0.01, $m$ = 0.017, $P < 0.0001$) as a function of *cis*-regulatory divergence (allele-specific expression in hybrids, ASE). Color scale indicates density of points with brighter colors denoting higher density.

recombination landscape, we asked whether distinct chromosomal domains would also associate with the degree of *cis*-regulatory divergence. We find higher molecular divergence between the genomes of *C. briggsae* and *C. nigoni* in chromosomal arms compared to centers in noncoding sequences upstream of protein-coding genes (S7A Fig), in addition to protein-coding sequence divergence (S8 Fig; also see [48]). These observations are consistent with the idea of stronger purifying selection on mutations to genes and their *cis*-regulatory regions when linked to chromosome centers, or more effective positive selection when linked to chromosome arms. Despite the elevated molecular divergence in arm regions, we only found modest elevation of ASE divergence for genes on arms (S7B Fig), as well as modest but significant differences in the magnitude of regulatory divergence in *cis* and *trans* between chromosome arms and centers (S7C Fig; *cis*-only GLM [females], coeff = 0.06, $t$ = 1.99, $P$ = 0.04; *trans*-only GLM [females], coeff = 0.05, $t$ = 4.45, $P < 0.001$). Notably, we observed only a weak positive correlation between ASE divergence and rates of molecular evolution (Fig 7A and 7B; linear regression for $K_a/K_s$': adjusted $R$ = 0.035, $m$ = 0.019, $P < 0.0001$; 1-$P_{cons}$: adjusted $R$ = 0.01, $m$ = 0.017, $P < 0.0001$). Overall, these patterns indicate that rates of divergence for gene expression and their *cis*-regulatory controls are largely decoupled from protein-coding sequence evolution.

## Discussion

Regulatory control over gene expression is an important component of phenotypic evolution [12]. As species diverge and accumulate mutations, selection will permit regulatory changes that maintain transcript levels as well as changes that allow exploration of new phenotypic space when they confer a fitness advantage. Sexual selection and sexual conflict can further promote such genomic divergence, both in terms of molecular evolution (e.g., rapid coding or regulatory sequence evolution for male-biased genes) and in terms of gene expression (e.g., divergence in sex-biased gene expression levels) [38,69,70]. In interspecies hybrids, sexually driven sources of genomic divergence can disrupt gene networks to create negative epistatic interactions that manifest as sex-biased hybrid sterility or inviability and generate reproductive isolation [5]. Here, we document extensive regulatory divergence in the face of both conserved

and divergent gene expression, with prominent influences of sex-biases and genomic location on the potential to induce misexpression in interspecies hybrids.

## Compensatory regulatory evolution implicates pervasive developmental system drift

*C. briggsae* and *C. nigoni* acquired substantial divergence at the DNA level since they diverged from their common ancestor ~35 million generations ago (3.5 Mya assuming 10 generations per year), including ~20% sequence divergence for synonymous sites, changes to genome size, and disproportionate loss of short male-biased genes in *C. briggsae* since its transition in reproductive mode to androdioecy [48,56]. Despite this genomic divergence, hybridization between these species yields viable and fertile $F_1$ hybrid females, as well as viable hybrid males that suffer complete sterility [36,49].

Despite observing substantial expression divergence, we nevertheless find that 39% of genes expressed in "females" (4,783 genes) and 46% in males (6,236 genes) show no differential expression between species. Conserved expression between species may result from stabilizing selection, recognized as a common force acting on transcript abundance [7–9,11]. Mechanistically, conservation of the expression phenotype can occur, despite sequence evolution, with co-evolutionary changes to both *cis-* and *trans*-regulatory elements. For example, if a *trans*-acting mutation fixes due to a pleiotropic benefit on other loci, then selection would favor fixation of any subsequent compensatory mutation in *cis* that returns expression to optimal levels at the focal locus [5,20]. We find evidence of widespread compensatory *cis-trans* divergence in gene regulation between *C. briggsae* and *C. nigoni*. Such coevolution represents just one mechanism leading to "developmental system drift," in which the molecular controls over developmental pathways can diverge while resulting in little or no change to their phenotypic outputs [14,21,74]. In *Caenorhabditis* nematodes, developmental system drift and stabilizing selection have been invoked to explain the high degree of phenotypic stasis and morphological constraint among species [60,75–78]. Gene network conservation despite *cis*-regulatory divergence has been demonstrated by inter-species promoter swaps in *Caenorhabditis*, showing both robustness in regulatory networks and neo-functionalization in specific cell types [20,78,79], as well concerted action of *cis* and *trans* compensatory regulation between more distantly related *Caenorhabditis* species [20]. Our results reinforce this view of pervasive developmental system drift: we show a high incidence of transgressively expressed genes (overdominant and underdominant in hybrids) and *cis-trans* compensatory changes in the sexually-dimorphic regulatory evolution and expression inheritance of each sex (**Fig 4**), in addition to an abundance of transgressive hybrid expression and *cis-trans* compensatory changes among genes with conserved sex-neutral expression profiles (C-N-N) (**Fig 5A–5D**).

Sequence divergence and developmental system drift in regulatory pathways can lead hybrids to experience misregulation due to the conflicting regulatory signals from the divergent genomes, yielding misexpression in hybrid transcriptomes [28]. This situation could present a Dobzhansky-Muller incompatibility if the misexpression leads to reduced fitness; genetic interactions like those experienced by hybrids have been untested by natural selection and will likely be detrimental [1]. The most striking signal of misexpression in our hybrids is the sharp contrast in the fraction of sex-biased genes: ~83% in hybrids vs ~60% in each parental species (**Fig 1C**). The degree of sex-biased gene expression is more extreme for the X chromosome in $F_1$ hybrids due in part to transgressive underdominance effects (**Figs 1D, 2C and 2D**), though the fraction of male-biased genes likely involved with spermatogenesis is, in fact, depleted on the X (**Figs 5G, 6C, 6E and 6G**), consistent with previous findings [43,44]. The X-chromosome in hybrid males is enriched, however, for genes showing overdominant expression, which can

be linked directly to *cis-trans* compensatory changes (**Figs 2D, 3 and S2**). Overall, these results implicate extensive sex-limited developmental systems drift that generates extensive misexpression of genetic networks distinctly in each sex for hybrids of *C. briggsae* and *C. nigoni*.

While expression differences between species are often biased towards one species in systems such as fruit flies and plants [16,18,25], we do not observe much asymmetry toward one species (**Fig 1A**; binomial test: males, ratio = 0.5, *P* = 0.26; females, ratio = 0.47, *P* < 0.001). This symmetry in expression suggests that demographic effects that exacerbate genetic drift are not likely to bias regulatory changes toward either increased or decreased expression disproportionately for one species, as could occur if regulatory changes fix more rapidly in species like *C. briggsae* with lower effective population sizes due to selfing.

## Sex differences in regulatory divergence expose sexual dimorphism of genetic networks

Transgressive expression in $F_1$ hybrids beyond the bounds of parental expression levels is a signature of misexpression, which we observe in abundance. Given that *C. briggsae* × *C. nigoni* hybrids obey Haldane's rule [49–51], we expected more misexpression in hybrid males. Our analyses revealed, however, that it is hybrid females that experience more extensive transgressive gene misexpression (**Fig 2A–2C**). Studies in mice have shown that hybrid misexpression of X-linked genes confers male sterility [5,29] and *Drosophila* demonstrates an important influence of X-autosome incompatibilities for Haldane's rule and hybrid male sterility [37,39], though with a more tenuous link to misexpression of X-linked genes [30,80]. We found enrichment of misexpressed loci on the X-chromosome for both sexes (**Fig 2D**) and no indication of compromised dosage compensation. Most female-biased transgressive genes show overdominant misexpression, however, compared to more underdominant misexpression in male-biased networks among hybrid males (**Fig 5A–5C**). This observation, in addition to higher divergence in expression and coding sequences (**Fig 5E and 5F**), suggest that male-biased networks are more "fragile" or prone to regulatory perturbations. Moreover, we hypothesize that *cis-* and/or *trans*-regulatory changes acquired after speciation experienced selection to sustain upregulated expression of female-biased genes, with those changes behaving in an overdominant manner in hybrids. Indeed, overdominant genes with *cis-trans* divergence in females have disproportionately evolved "enhancing" regulatory changes compared to males (**Figs 2G, 3 and 5D**). In spite of the extensive overdominance in female-biased expression (**Figs 2B, 2C and 5C**), the fact that hybrid females are fertile suggests that overdominant expression does not impact fitness as negatively as does regulatory divergence that leads to underdominance in hybrids. Consequently, this sexual dimorphism in gene regulatory networks may contribute to the greater sensitivity of males to manifesting hybrid dysfunction.

Interestingly, we find that regulatory controls over male-biased genes when they are expressed in $F_1$ females largely reveals *trans*-only regulatory changes (**Figs 5D, 5E, 6F and 6G**). These *trans*-acting regulatory changes are more strongly associated with *C. nigoni* dominant expression in females among autosomes, which contrasts with these same genes when expressed in males that reveal extensive *cis*-regulatory changes and *C. briggsae* dominant expression (**Figs 4, 5C and 5D**). Previous studies have shown that *trans*-acting regulation often manifests expression dominantly in hybrids, possibly as a result of masking effects between dominant and recessive *trans* alleles, whereas *cis*-acting regulation often generates additive contributions in expression [64]. However, *cis*-dominant expression patterns are not uncommon in ASE studies [18] and may arise, potentially, from cases where *cis* changes in one species decrease transcription factor binding affinity or where significant post-transcriptional regulation in pure species is compromised in hybrids (see Results section). Merritt et al. [65]

showed that *C. elegans* oogenic germline gene networks depend strongly on 3'UTR post-transcriptional regulation, but that genes showing spermatogenic expression rely primarily on upstream transcriptional regulation. Curiously, disruption of endogenous 22G small RNA regulation of spermatogenesis in hybrids of *C. briggsae* x *C. nigoni* also is implicated in male sterility [31]. It may be that sex differences in the dominant mode of regulation (transcriptional versus post-transcriptional) contributes to the sex differences in the *cis*-dominance that we report. These results also align with observations of downregulation of spermatogenesis genes, such as *fog-1* and *fog-3*, by specific transcription factors (i.e., *tra-1*), and sperm-specific expression depending more on upstream promoter regions than 3-UTRs in *C. elegans* [65,72,81].

We observed distinct signatures of regulatory divergence for genes with sex-biased expression when expressed in the opposite sex, raising the possibility that such divergence may reflect a history of sexual selection and sexual conflict [52,57]. One possible explanation invokes Rice's hypothesis [82]: the fixation of regulatory changes that create sex-biased expression serve as evolutionary solutions to inter-sexual genomic conflict. To avoid traits that are detrimental to females but improve male performance, genomic conflict resolution by means of sex-biased expression may be attained faster through *trans*-regulatory changes, which are more pleiotropic, downregulating male-biased genes in females. This logic aligns with the hypothesis that sex-biased expression is partly driven by selection acting to resolve sexual conflict by means of modifier alleles or regulators [67,82]. However, the fact that *trans*-only regulatory changes do not predominate in the control of female-biased genes in males (**Fig 4E**), suggests that regulatory mechanisms to resolve genomic sexual conflict act in different ways for the two sexes. Direct manipulative experiments would prove valuable in testing these ideas further.

The fact that the egg-bearing sex in the *C. briggsae* parent is actually a hermaphrodite may help explain the presence of underdominant genes in hybrid females. Many of the genes in hybrid females that show underdominant effects would otherwise show male-biased expression (**Figs 5C and 6E**), suggesting that they may compromise spermatogenesis to effectively convert $F_1$ hermaphrodites into females. This perspective provides a complementary view to the idea that hermaphroditism is 'recessive' to femaleness in a simple Mendelian manner [49]. Our expression data characterizes *tra-1*, a master regulator of sexual development programs through its repression of genes such as *fog-3* that would promote spermatogenesis and male development [72], as a conserved sex-neutral gene that is overdominant in both male and female $F_1$ hybrids (C-N-N; **S6 Fig**). Furthermore, *tra-2* and *fem* genes, which interact to facilitate TRA-1 activity as a repressor, are all female-biased and overdominantly expressed in hybrids, which suggest an additional mechanism for hybrid XX "demasculinization". Consequently, repression of male-biased genes involved with the *tra-1* pathway is likely to cause both the "female" phenotype in hybrid females (i.e. absence of male function normally seen in hermaphrodites) and reduced fertility in hybrid males.

Gene expression in hybrid males predominantly shows either simple dominance or no change (**Fig 2C**). While it is tempting to speculate that it might be a byproduct of males of different species sharing the same reproductive role, the combined observations of reduced sexual selection in *C. briggsae* males [53,57], genomic divergence [55,56], and clear sex differences in hybrid fertility [49,50], suggest otherwise. If most transgressive expression occurs in the gonad, then the small and defective gonad development of $F_1$ males may have led to their observed paucity of transgressive expression. Distinct relative sizes of tissues between species, sexes, and hybrids could potentially influence differences in expression and their interpretation [83]. Several aspects of these *Caenorhabditis*, however, minimize the potential influence of allometric effects between species. The morphology of *Caenorhabditis* species is highly conserved even between very distantly related species, with body size being very similar for *C.*

*briggsae* and *C. nigoni* in particular [84]; body size differences within and between species reflect differences in cell size not cell numbers [59]. Our data show symmetry in upregulated genes between species (**Fig 1A**), consistent with a limited potential role of allometric bias between species. While hybrid females experience delayed germline development [49], we analyzed adult animals with complete development. However, the gonad of hybrid males is anatomically deformed, potentially contributing to differences in expression due to tissue allometry. If disproportionately small gonad size of hybrid males were to skew expression patterns, then we would expect a large signal of genes showing low expression specifically in hybrid males. We observe two modules of co-expression with such a pattern (M3 and M15 in **S2 Fig**), but M15 comprises just 437 (8.7%) of male-biased genes and M3 contains genes with female-biased expression overall (672 genes, 15% of female-biased genes). Moreover, we observe a high number of genes with conserved expression across males, including hybrids (**Fig 2C**), counter to what would be expected if tissue allometry introduced substantial bias.

## Implications for Haldane's rule and the large-X effect

Male-biased genes are expected to evolve fast because of sexual selection and sexual conflict, resulting in higher rates of protein-coding and gene expression divergence [38,69,70]. Faster evolution of male-biased genes is the premise behind the "faster male" hypothesis to explain the high incidence of hybrid male sterility in Haldane's rule [33]. We find faster evolving coding sequences for the subset of male-biased genes that also show exceptionally high expression divergence (B-M-I and B-M-N; **Fig 5E and 5F**), many of which are implicated in spermatogenesis (**Figs 6A–6C**; **S4 and S5**). This result is consistent with previous reports of faster molecular evolution of spermatogenesis and male germline genes of *C. elegans* [57,85,86], and suggests that "faster male" evolution may contribute to Haldane's rule in *Caenorhabditis*. To the extent that rapid evolution predisposes genes to forming genetic incompatibilities, however, the fact that we rarely observe such genes on the X-chromosome suggests that the "faster X" model does not provide a compelling explanation for Haldane's rule for hybrid male sterility [34] (**Fig 5G**).

The "large-X" effect, where hybrid male sterility results from a disproportionate count of X-linked loci involved in genetic incompatibilities [87], is evinced from deletion screen experiments in *Drosophila* [37,39] and introgression experiments in *C. briggsae* x *C. nigoni* [58,59]. An analogous effect can result from X-linked regulatory divergence that causes misexpression and hybrid dysfunction [26,29,88]. Consistent with a large-X effect due to regulatory evolution in *Caenorhabditis*, our analyses show that the X chromosome contains an excess of genes that have undergone *cis-trans* compensatory changes causing misexpression in hybrid males (**Fig 3**). This pattern implicates a disproportionate role for regulatory divergence of the X-chromosome in mediating misexpression in *Caenorhabditis* hybrids.

Two non-mutually exclusive ways in which hybrid male dysfunction (i.e., sterility) can arise are: 1) through misexpression and misregulation of X-linked genes involved with male function, and 2) through negative epistatic interactions (i.e., incompatibilities) between X-linked and autosome genes involved in male-specific pathways. Our results suggest that both cases are plausible.

First, the paucity of X-linked sex-biased genes in parental genotypes of *Caenorhabditis* species suggests that any misregulation and misexpression on the X might exert little downstream impact (**Fig 1D**; [43,44]). However, misexpression of X-linked genes in hybrids is relatively common compared to autosomes (with the exception of Chromosome V) (**Fig 2D**), with hybrid males having higher relative incidence of effectively misregulated genes (*trans*-only and compensatory *cis-trans* changes) compared to female hybrids (**Fig 3**). Although enrichments

on the X were non-significant, we find other signs of *trans*-acting factors contributing to misexpression in both sexes (**Figs 3 and 4**). In hybrid females, *trans*-acting factors largely drive the expression of X-linked genes with *C. briggsae* dominant and underdominant expression, unlike autosomes (except Chromosome V) (**Fig 4**). In hybrid males, our inference that all X-linked genes with *C. briggsae* dominant expression arise from *trans*-only* effects implicating recessive *C. nigoni trans* regulators (**S13 Fig**), and which, at minimum, should be an underestimation of all *trans*-acting regulatory divergence affecting the X-chromosome, suggests that similar *trans*-acting misregulation occurs. These findings are consistent with previous observations, particularly in *Drosophila*, of *trans*-acting sex-specific changes causing misregulation of X-linked genes [40,89,90].

Second, the extensive expression dominance in $F_1$ males that disproportionately matches the *C. briggsae* expression level due to *cis* effects have the potential to disrupt gene networks as they may interact negatively with *C. nigoni* X-linked genes in hybrid males (**Figs 4 and S9**). Autosomal spermatogenesis genes, by contrast, tend to show *C. nigoni*-dominant expression in $F_1$ hybrid males (**S5B and S5C Fig**), consistent with previous work showing recessive effects of the *C. briggsae* autosomal portions of genetic incompatibilities [58]. In addition, this prior work also showed that sterility in *C. nigoni* x *C. briggsae* hybrid males may not require many X-autosome incompatibilities [59]. Despite their low abundance on the X-chromosome, X-linked spermatogenesis genes are often enriched for both misexpression and misregulation (**Figs 3, S5B and S5C**), potentially enhancing their role in hybrid dysfunction. Our genome-wide transcriptome analysis of *cis*- and *trans*-regulatory divergence therefore sheds new light on Haldane's rule, reinforcing some previous key inferences about hybrid dysfunction associated with males, spermatogenesis, and the X-chromosome.

### Decoupled coding vs regulatory divergence and the evolution of hybrid dysfunction

Genes that evolve fast may be predisposed to creating genetic incompatibilities, which could result either from dysfunctional structural activities of the encoded proteins or controls over the timing or location of gene expression. Studies to date indicate that rates of evolution of coding sequences and regulatory regions do not correlate strongly [91,92]. Supporting the idea that regulatory divergence due to *cis*-acting elements is largely decoupled from rates of molecular evolution, we found only weak positive correlations between rates of molecular for protein structure and gene regulation (**Fig 7**). *cis*-regulatory divergence also showed only a weak elevation in chromosome arms compared to centers (**S7B Fig**), genomic regions with marked differences in recombination rates, gene density, sequence conservation within and between species (**S7A and S8 Figs**) [48,93] that influence the rate at which mutations, especially weakly-selected regulatory mutations, can get fixed as a result of direct selection and linked selection [94]. The decoupled rates of evolution for protein structure and gene regulation imply that genetic incompatibilities mediated through protein activity versus gene regulation may follow different rules in the evolution of reproductive isolation.

### Conclusion

We contrasted sex-specific transcriptomic profiles between *C. briggsae* and *C. nigoni* and their hybrids to understand how the evolution of *cis*- and *trans*-regulatory elements can contribute to $F_1$ hybrid dysfunction. Regulatory evolution underlies divergent expression as well as conserved expression subject to compensatory effects of changes to multiple elements. The sharp contrast of extensive misexpression in $F_1$ hybrids with the morphological stasis and expression conservation between *Caenorhabditis* species indicates substantial developmental system drift

of regulatory networks that destabilize in hybrids to enforce reproductive isolation between species. Despite more extensive transgressive expression in hybrid females, they are fertile but unable to produce self-sperm and hybrid males are entirely sterile, suggesting that 1) genetic networks controlling "male" developmental pathways are more fragile in the face of genetic perturbation and 2) hybrid females may represent "demasculinized" hermaphrodites through the disruption of sperm-specific regulatory networks. Despite the rarity of sex-biased genes on the X-chromosome, the X is home to disproportionate misexpression in both sexes, with misregulation in hybrid males largely occurring through *cis-trans* compensatory changes, but also by *trans*-acting factors to some extent. X-autosome incompatibilities in hybrid males likely result from the propensity for divergent *cis*-acting factors to drive *C. briggsae*-dominant autosomal expression yielding allele-specific expression biases, which then interact negatively with *C. nigoni* X-linked genes. Moreover, *C. nigoni*-dominant *trans*-acting factors may act to downregulate male-biased genes in both males and females, through the misregulation of master controllers of sexual development such as *tra-1*. Finally, we find only weak correlations of *cis*-regulatory divergence with chromosome architecture and protein-coding and non-coding sequence divergence, indicating that regulatory and protein evolution are largely decoupled. Consequently, Dobzhansky-Muller incompatibilities involving regulatory and coding sequences may accumulate independently of one another, and in distinct ways in the regulatory networks of each sex, building-up reproductive isolation that leads to Haldane's rule and speciation.

## Material and methods

### Samples, RNA isolation, and sequencing

We cultured triplicate populations of isofemale *C. briggsae* (AF16) and *C. nigoni* (JU1421) on NGM-agar plates with *Escherichia coli* OP50 at 25˚C, isolating total RNA via Trizol-chloroform-ethanol extraction from groups of approximately 500 individual age-synchronized young adult males or females (hermaphrodites) for each replicate sample. *C. briggsae* hermaphrodites are treated as the female sex for the purposes of this study (see below and the Results section), as their soma is phenotypically female despite the gonad producing a small number of sperm in addition to abundant oocytes. We also crossed in triplicate virgin *C. nigoni* females to male *C. briggsae* (isolated as L4 larvae) to produce $F_1$ hybrid progeny, with RNA isolated from male and female $F_1$ hybrids as for the parental pure species genotypes.

The triplicate mRNA samples for each sex and genetic group (*C. briggsae*, *C. nigoni*, $F_1$ hybrid) underwent 100bp read length, single-ended Illumina HiSeq sequencing at Genome-Quebec according to their standard TruSeq3 protocol. A total of ~250 million reads from these 18 barcoded samples spread across 4 lanes were cleaned for quality control using Trimmomatic v0.38 (with arguments: ILLUMINACLIP:TruSeq3-SE:2:30:10 LEADING:3 TRAILING:3 SLIDINGWINDOW:4:15 MINLEN:36) [95].

### Reference alignment and allele-specific read assignment

Following quality control trimming and filtering, we mapped sequence reads from each sample to the chromosome-level genome assembly and annotation of each species (*C. briggsae* WS271 https://osf.io/a4e8g/, *C. nigoni* 2018-01-WormBase https://osf.io/dkbwt/; [56]) using STAR v2.6 (https://github.com/alexdobin/STAR; [96]) with default parameters and adjusting for intron size (—alignIntronMin 40—alignIntronMax 15600). Reads from *C. briggsae* and *C. nigoni* were mapped to their own reference, while reads from $F_1$ hybrids were mapped to both reference genomes. Reference genomes and GFF files can also be found online (https://github.com/santiagosnchez/competitive_mapping_workflow/tree/master/references).

To obtain allele-specific read counts in $F_1$ hybrids, we applied a competitive read mapping approach by developing a Python implementation for competitive read-mapping we call COMP-MAP (https://github.com/santiagosnchez/CompMap). This approach uses the PYSAM library (https://github.com/pysam-developers/pysam) internally for reading and indexing BAM alignments by read name. Using BED files with transcript-level coordinates consistent between references, read overlaps were then counted for each feature in both alignments. At each feature, the alignment score (AS) and number of mismatches (nM) of each read to both references were compared, assigning best aligned to reference-specific counts. Ambiguous reads (i.e., those having equally good alignments) were also counted and redistributed proportionally to the number of reads assigned to each reference. We validated our method with simulated RNA-seq data using the R package POLYESTER [97] finding high correlation between true ASE and COMPMAP counts, as well as low type 1 and type 2 error rates among *cis* and *trans* regulatory divergence categories (<5%; **S10** and **S11** Figs). We expected to have high power to detect ASE, given ~20% neutral sequence divergence between *C. briggsae* and *C. nigoni* [48] conferring on average ~5 nucleotide differences for every 100 bp of coding sequence (0.2 divergence * 0.25 fraction of synonymous sites * 100 bp). Additionally, we did not expect significant mapping bias [98,99] given that our data was strain-specific (i.e. our C. *briggsae* RNA-seq data comes from strain AF16, which is the same as the reference genome and our *C. nigoni* data comes from strain JU1421 which derives from the same wild isolate as the *C. nigoni* reference genome; see supplementary data in [56]). Scripts, programs, and commands used for bioinformatic analyses can be found online (https://github.com/santiagosnchez/competitive_mapping_workflow).

## Ortholog identification and read abundance quantification

The chromosomes in the *C. briggsae* and *C. nigoni* genomes are largely colinear, with only a few small inversions and translocations reported [100]. Therefore, we quantified gene expression abundance for a set of 13,636 genes that we inferred to be one-to-one reciprocal orthologs between *C. briggsae* and *C. nigoni*, for which upstream regulatory regions should also be syntenic. To identify orthologs, we applied a phylogenetic approach using ORTHOFINDER v2.2.6 [101,102], based on longest-isoform peptide sequence translations for gene annotations of 28 *Caenorhabditis* species [103] (http://caenorhabditis.org/). BLASTp all-by-all searches were done separately on SciNet's Niagara supercomputer cluster. ORTHOFINDER was run with default options, which included: -M dendroblast (gene tree reconstruction) and -I 1.5 (MCL inflation point). In further analysis of the final set of 13,636 orthologs, we excluded from a preliminary set of 15,461 orthologs those genes for *C. briggsae* and *C. nigoni* that could not be assigned to any of their six chromosomes (688 genes), that were associated with inter-chromosomal translocations (370 genes), that we could not estimate $K_a/K_s$ reliably (275 genes), or that exhibited low mRNA-seq read abundance (492 genes; see below). A list of all the orthologs can be found online (https://github.com/santiagosnchez/competitive_mapping_workflow/blob/master/orthologs.txt).

We quantified gene expression in parent species for each ortholog with FEATURECOUNTS v2.0.1 [104] as we found its read-counting method to be compatible with COMPMAP. Raw read counts were combined into a single table and imported into R together with ASE counts [105] for normalization and statistical analyses. All raw data counts can be found online (https://github.com/santiagosnchez/competitive_mapping_workflow/tree/master/counts).

## Identification of male-biased genes in *C. briggsae* hermaphrodites

*C. briggsae* hermaphrodites have the ability to produce low amounts of sperm in addition to oocytes, while their soma phenotypically resembles that of a female. Therefore, to make more

biologically realistic contrasts between *C. briggsae* hermaphrodites, *C. nigoni* females, and female hybrids, we used an *in silico* approach to identify male-biased genes, which are also upregulated in hermaphrodites, through comparisons using previously published RNA-seq data. We used RNA-seq read data derived from *C. briggsae* pseudo-females (AF16-derived *she-1(v47)* mutant strain) [54], which are unable to produce self-sperm. We conducted differential expression analyses using a similar pipeline as described below to identify genes with significant sex-biased expression in our *C. briggsae* RNA-seq data and the one from pseudo-females in Thomas et al. [54], in addition to being differentially expressed between datasets. These genes where then excluded from analyses comparing hermaphrodites and females together and were analyzed separately. Data for these analyses can be found online (https://github.com/santiagosnchez/competitive_mapping_workflow/tree/master/analyses/tables/Thomas_et_al_data).

## Differential expression analyses: Contrasts between species, hybrids, and sexes

We used the R Bioconductor package DESEQ2 [106] to assess differential expression. Before statistically assessing differential expression, we summed the allele-specific counts from $F_1$ hybrids to yield a single count of transcripts per gene. After calculating library size factors, we filtered out genes that did not meet the criterion of having at least 3 samples with more than 10 library-size scaled counts. We visualized the overall expression distance between samples using a non-metric multi-dimensional scaling plot, which showed all three biological replicates to cluster consistently within their corresponding treatment (**S1A Fig**). We inferred sex-biased gene expression by comparing differential expression profiles between males and females (or hermaphrodites) in each genetic group (*C. briggsae*, *C. nigoni*, $F_1$ hybrids). We also quantified differential expression between the genetic groups in a pairwise manner (*C. briggsae* vs $F_1$, *C. briggsae* vs *C. nigoni*, *C. nigoni* vs $F_1$) for each sex separately. We then contrasted expression patterns between species (*C. briggsae* and *C. nigoni*) by looking at sex differences (sex-biased expression) and their interaction (expression ~ species * sex). Within hybrid males and females, we compared allele-specific counts to measure ASE. For all of these contrasts, we used the negative binomial generalized linear model fitting and Wald statistics to determine differentially expressed genes, as implemented in DESEQ2. We used FDR-adjusted *P*-values at the 5% level to assess significance [107]. DESeq2 results can be found online (https://github.com/santiagosnchez/competitive_mapping_workflow/tree/master/analyses/tables/DESeq2).

## Co-expression clustering

To identify groups of genes with shared co-expression trends, we averaged $\log_2$-transformed normalized read counts with the *rlog* function from DESEQ2 for each sample. Then we standardized gene-wise expression data by calculating Z-score values and used *K*-means to cluster co-expression groups. We chose a sensible *k* value (*k* = 15) approaching the asymptote by plotting the within-group sum of squares for a range of *k* values (from 2 to 100). We then calculated centroid expression levels by estimating the mean relative expression across samples within each group. Co-expression modules were designated as M1-M15. Our co-expression clustering results can be found online (https://github.com/santiagosnchez/competitive_mapping_workflow/tree/master/analyses/tables/clustering).

## Mode of expression inheritance in $F_1$ hybrids

Based on patterns of expression in $F_1$ hybrids relative to parent species, we classified genes into those having **additive** (intermediate), **dominant** (matching either of the species), **overdominant** (higher that both parents), and **underdominant** (lower than both parents) profiles following

the logic in McManus et al. [16]. Genes with no significant differences in expression between F₁s and their parent species were deemed to have conserved regulatory controls resulting in **no change** in expression in F₁s. Genes with additive effects had intermediate expression in F₁s compared to both parental species, meaning that there were significant differences in expression between all groups in a manner where expression levels in F₁s fall in between both species. Genes with dominant effects were those with expression levels in F₁s matching either one of the parent species (i.e., no significant differential expression), but with significant differential expression between species. Finally, genes with significant differential expression from both parents, but that were either significantly underexpressed (underdominant) or overexpressed (overdominant) compared to both species were regarded as transgressive. Genes falling outside any these specific categories were considered **ambiguous**. A per-gene summary table with all expression inheritance classification can be found online (https://github.com/santiagosnchez/competitive_mapping_workflow/tree/master/analyses/tables/expression_inheritance).

We also measured absolute Euclidean distances in expression relative to the centroid or origin in expression space of F₁ hybrids relative to both parent species. For example, for every gene we took the expression difference between F₁s and *C. briggsae* and between F₁s and *C. nigoni* as an xy coordinate system. Then, we measured the Euclidean distance from that point in expression space to the origin (0,0), reflecting no change in expression:

$$d = \sqrt{\left(\Delta_{F1/Cbr}\right)^2 + \left(\Delta_{F1/Cni}\right)^2}$$

Where $\Delta_{F1/Cbr}$ and $\Delta_{F1/Cni}$ are coefficients of differential expression between F₁ hybrids and each parent species. This metric allowed us to visualize the magnitude of expression distance from a "conserved" expression profile.

### *cis*- and *trans*- regulatory divergence

We used ASE in F₁s to quantify the extent and type of *cis*- and *trans*-regulatory differences between species. Expression divergence between parent species results from both *cis*- and *trans*-regulatory changes, whereas significant differential expression between alleles in F₁s results from *cis*-regulatory divergence only [16]. To quantify the extent of *trans* effects, we applied a linear model to test for differences in gene expression between parent species (P) and between alleles in F₁ hybrids (ASE) using the following model: expression ~ species/group, where "group" represented categorical variables pointing to data from P and ASE. The division operator of the function "/" measures expression ratios independently for each category in "group". We then used a post-hoc Wald-type test (*linearHypothesis* from the CAR package) to test for significant differences between both coefficients (**P**[*C. nigoni*/*C. briggsae*] = **ASE**[*C. nigoni*/*C. briggsae*]). *P* values were considered significant after a 5% FDR analysis [107].

We inferred the influence of *cis*- and *trans*-regulatory divergence on genes linked to autosomes, as well as to the X-chromosome in females, following the criteria in McManus et al. [16]. This procedure allowed us to designate genes having undergone significant regulatory divergence due to **cis-only**, **trans-only**, and **cis-trans** effects. Genes with significant *cis* and *trans* effects were split into those having synergistic effects or **cis + trans** and those having compensatory effects: **cis x trans (compensatory)** and **cis-trans (compensatory)** (S12 Fig). Genes expressed with no significant differences in expression between parents, ASE, or *trans* effects were deemed as **conserved** and those that did not strictly fit into any of the previous groups were considered **ambiguous**. A table with all gene-wise classifications for expression inheritance in all chromosomes for females and for autosomes in males can be found online (https://github.com/santiagosnchez/competitive_mapping_workflow/tree/master/analyses/tables/cis_trans).

Given the hemizygous condition of the X-chromosome in males, we cannot use $F_1$ ASE of X-linked genes to assess *cis* and *trans* regulatory divergence. However, we devised a scheme to assign different types of regulatory divergence to X-linked genes, with some limitations, based on the differences in expression between male $F_1$ hybrids and parent species (**S13 Fig**). $F_1$ males in our study carry their maternal *C. nigoni* X-chromosome. Therefore, assuming that the bulk of the *trans* environment derives from autosomes, we would expect that X-linked genes that differ in expression between parental species but display *C. nigoni* dominant expression owe their regulatory divergence mostly to *cis*-regulatory (**cis-only**∗) changes, because any significant deviation from *C. nigoni* expression levels would indicate *trans*-regulatory changes. Potentially confounding situations would involve (1) the action of "local" *trans* regulators found on the X, albeit still considered "local", and (2) autosomal *trans* regulators that are dominantly expressed by the *C. nigoni* allele. Given that the confounding effects of the dominant expression of *trans* regulators would also potentially apply to autosomal genes, we decided to keep the *cis*-only∗ category for X-linked genes. Moreover, genes with significant regulatory divergence between species where hybrids display *C. briggsae* dominant expression (**trans-only**∗) would indicate no significant changes due to *cis* regulation, with the condition that autosomal *C. nigoni trans* regulators affecting those genes are recessive, and therefore not expressed. We consider these a subset of total *trans* regulation and is likely an underestimation compared to autosomes and the X-chromosome in females. The last category that we can assign confidently is **compensatory *cis-trans* changes**. Genes with no differential expression between species, but with significant up- or down-regulation in F1 males (i.e., overdominant or underdominant) were considered as having *cis-trans* compensatory changes. For the purpose of comparing regulatory divergence between autosomes and the X in males and females, we lumped all other genes with significant expression divergence into "other", as they would be difficult to disambiguate without additional data. These include genes with *trans*-only (i.e., with codominant *trans* regulation), *cis* + *trans* (enhancing), *cis* x *trans* (compensatory), and ambiguous. Genes with no expression change were kept as "conserved" (**S11 Fig**).

## Molecular evolution in coding sequences

Orthologs in the genomes of both *C. briggsae* and *C. nigoni* were first aligned as protein coding sequences using MAFFT v7.407 [108]. These alignments were then back-translated to coding sequence (CDS) alignments using the python program CODONALIGN (https://github.com/santiagosnchez/CodonAlign). We estimated rates of synonymous site ($K_s$) and non-synonymous site divergence ($K_a$) between the two aligned sequences using a custom Python script (https://github.com/santiagosnchez/DistKnKs) applying the Yang and Nielsen (2000) model implemented in BIOPYTHON. We also corrected $K_s$ values for selection on codon usage using the effective number of codons (ENC; [109,110]) as a predictor in a linear model. In short, we fitted a linear regression model ($K_s$ ~ ENC), which we used to predict $K_s$ at the maximum value of ENC (= 60). Then, we corrected the bias in $K_s$ by adding the residuals of the linear model to that idealized value of $K_s$ at ENC = 60. We refer to these corrected set of $K_s$ estimates as $K_s$'. A table with these estimates of molecular evolution for each gene can be found online (https://github.com/santiagosnchez/competitive_mapping_workflow/tree/master/analyses/tables/molecular_evolution).

## Upstream non-coding sequence conservation

Chromosome-level FASTA sequences for *C. briggsae* and *C. nigoni* were aligned using LASTZ [111], outputting alignment files for each chromosome in MAF format. We used BEDTOOLS's v2.27 [112] *flank* function to generate 500 bp intervals of the 5' upstream flanking regions of

each orthologous gene. We then used *maf_parse* from PHAST [113] to extract overlapping alignment blocks of at least 500 bp long. We quantified sequence conservation as the average number of identical 5 bp non-overlapping windows between aligned DNA in both sequences.

## Spermatogenesis genes

To infer genes involved with spermatogenesis, we downloaded a list of *C. elegans* genes previously identified as spermatogenesis-related based on tissue-specific transcript abundance [45] (Additional File 4). We then used the BioMart tool of the WormBase Parasite website [114] to retrieve *C. briggsae* orthologs from the list of *C. elegans* genes. We cross-referenced *C. briggsae* orthologs to our own set of orthologs between *C. briggsae* and *C. nigoni* and annotated the 1,089 gene matches with a spermatogenesis tag. The data obtained for *C. elegans* spermatogenesis orthologs can be found online (https://github.com/santiagosnchez/competitive_mapping_workflow/tree/master/analyses/tables/spermatogenesis).

## Supporting information

**S1 Fig. Distinct expression profiles encompass species, sexes, F1 hybrids.** (A) Multi-dimensional scaling plot showing overall expression distance between samples. (B) magnitude of expression (rlog-transformed counts) for autosomes and the X-chromosome across *C. briggsae* hermaphrodites, *C. nigoni* females, and F1 females. The X-chromosome has, on average, lower magnitude of gene expression than autosomes, which is expected with dosage compensation. However, F1 hybrids do not show a strong pattern of over-active dosage compensation leading to generalized lower levels of expression.
(TIF)

**S2 Fig. Centroid relative expression of co-expression clusters and chromosomal enrichments grouped by sex-biased expression.** The naming scheme for each of the clusters is arbitrary. Numbers inside circles indicate the number of genes in each cluster. Enrichment/depletion within chromosomes is represented by the $\log_2$ odds ratio (i.e., observed/expected), with positive values indicating enrichment and negative values depletion.
(TIF)

**S3 Fig. X-autosome differences in regulatory controls in females and between sexes for autosomes underlie hybrid transcriptomic profiles relative to parent species.** Heatmap of the number of genes in each expression inheritance group (x-axis) for each type of *cis* and *trans* regulatory changes (y-axis) for each chromosome (I-V, X) and each sex. Includes all categorizations.
(TIF)

**S4 Fig. Centroid of normalized expression and 95% CI of species-by-sex groups (see Fig 5) plotted as reaction norms.** Rows show relative expression of sex-neutral, male-biased, and female-biased genes. Groups that include an "I" at the end in the name code have significant species-by-sex interactions.
(TIF)

**S5 Fig. Male-biased, hemaphrodite upregulated genes (see Fig 6) in males have similar expression inheritance, (B) but not expression distance (A), to females (Fig 6).** The X-chromosome is depleted compared to autosomes (B) but has distinct relative enrichments of genes with *C. briggsae* expression dominance in both males and "females" in contrast to autosomes, and presents different misexpression categories between males (overdominant) and females

(underdominant).
(TIF)

**S6 Fig. Genes involved with sex determination and germline development have significant divergence in regulatory networks.** The left panel shows relative (log-transformed and normalized) expression for males (brown) and females (light green) in *C. briggsae* (Cbr), F$_1$ hybrids (HF1), and *C. nigoni* (Cni). The right panel shows a biplot between the log-fold change in allele-specific expression (y-axis) and the log-fold change in expression divergence between species (x-axis). Dotted lines mark expected trajectories for *trans*-only (horizontal), *cis*-only (diagonal), and *cis-trans* compensatory (vertical) changes. Grey dots represent the regulatory space of other orthologs in our dataset.
(TIF)

**S7 Fig.** Chromosomal arm (green) and center (purple) regions differ strongly in (**A**) upstream sequence divergence (1-$P_{cons}$; proportion of non-conserved 5 bp windows within 500 bp upstream of each gene) but only moderately in (**B**) *cis*-regulatory divergence (log$_2$ allele-specific expression, ASE; females on top). (**A**) Proportion of non-conserved 5 bp windows within 500 bp upstream of each gene (1-$P_{cons}$) for each chromosome. (**B**) Absolute magnitude of log$_2$ allele-specific expression or *cis*-regulatory divergence for each chromosome in females and for autosomes only in males. Black lines in A and B indicate general additive regression (GAM) trendlines. (**C**) Arm and center regions do not differ greatly in magnitude of regulatory divergence. Box- and density plots of total (absolute) *cis*-regulatory divergence (left two panels) and *trans* regulatory divergence (right two panels) for either male or female gene expression. Allele-specific expression values are square-root-normalized in C to facilitate visual comparison.
(TIF)

**S8 Fig.** Coding sequence evolutionary rates for replacement sites ($K_a$) and synonymous sites ($K_s$', adjusted for selection on codon usage) for 13,636 orthologs between *C. briggsae* and *C. nigoni* along the chromosome positions of the *C. briggsae* genome. Colors mark chromosome arms (green) and center (purple); black lines indicate general additive regression (GAM) trends.
(TIF)

**S9 Fig.** Proportion of autosomal genes with significant allele-specific expression (y-axis) showing either additive expression (first panel, top row), simple expression dominance of one species (second and third panels, top row) and transgressive over- or under-dominant expression (bottom panels) in hybrid males. Genes with conserved regulation or *trans*-only effects were excluded for not having significant allele-specific expression.
(TIF)

**S10 Fig.** Scatter plot and linear regressions (top 4 panels) between regulatory divergence (log-fold-change) due to allele-specific expression or *cis* changes and their FDR corrected *P*-values, regulatory divergence due to *trans* effects and their FDR corrected *P*-values, and volcano plots (lower panels) comparing results from simulated RNA-seq allele-specific expression data generated by *polyester* and allele-specific read counts by CompMap.
(TIF)

**S11 Fig. Type 1 (false positive) and type 2 (false negative) error rates for each category of regulatory divergence.** Expected values were drawn from RNA-seq allele-specific expression data simulated with *polyester*. Observed values were drawn based on inferences drawn from

CompMap allele-specific count data.
(TIF)

**S12 Fig. Qualitative examples of allele-specific expression with their classification of gene regulation type changes.**
(TIF)

**S13 Fig. Qualitative examples to diagram regulatory divergence scoring for the X-chromosome in males that are hemizygous for the X (i.e., X0).** Categories not fitting into either *cis*-only, *trans*-only, or compensatory *cis-trans* were lumped into "other".
(TIF)

## Acknowledgments

We thank Katja Kasimatis for comments on previous versions of the manuscript. Computational analyses were performed on the Niagara supercomputer cluster at the SciNet HPC Consortium.

## Author Contributions

**Conceptualization:** Santiago Sánchez-Ramírez, Jörg G. Weiss, Cristel G. Thomas, Asher D. Cutter.

**Data curation:** Santiago Sánchez-Ramírez.

**Formal analysis:** Santiago Sánchez-Ramírez.

**Funding acquisition:** Asher D. Cutter.

**Investigation:** Santiago Sánchez-Ramírez, Asher D. Cutter.

**Methodology:** Santiago Sánchez-Ramírez, Jörg G. Weiss, Cristel G. Thomas.

**Project administration:** Santiago Sánchez-Ramírez, Asher D. Cutter.

**Resources:** Jörg G. Weiss, Cristel G. Thomas, Asher D. Cutter.

**Software:** Santiago Sánchez-Ramírez.

**Supervision:** Asher D. Cutter.

**Validation:** Santiago Sánchez-Ramírez.

**Visualization:** Santiago Sánchez-Ramírez.

**Writing – original draft:** Santiago Sánchez-Ramírez.

**Writing – review & editing:** Santiago Sánchez-Ramírez, Asher D. Cutter.

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
