## [Decision Letter · Decision Letter 0]

22 Jan 2021

Dear Dr Sanchez-Ramirez,

Thank you very much for submitting your Research Article entitled 'Widespread misregulation of inter-species hybrid transcriptomes due to sex-specific and sex-chromosome regulatory evolution' to PLOS Genetics.

The manuscript was fully evaluated at the editorial level and by independent peer reviewers. The reviewers appreciated the attention to an important topic but identified some concerns that we ask you address in a revised manuscript

We therefore ask you to modify the manuscript according to the review recommendations. Your revisions should address the specific points made by each reviewer.

[LINK]

Yours sincerely,

Alex Buerkle

Associate Editor

PLOS Genetics

Bret Payseur

Section Editor: Evolution

PLOS Genetics

This revised manuscript has been reviewed by three referees, who concur that the revisions have substantially improved the manuscript and that only minor points remain to be addressed. I encourage the authors to consider these additional suggestions for improvement of the manuscript.

Reviewer's Responses to Questions

**Comments to the Authors:**

Reviewer #1: I think that the authors have addressed most of the issues I have raised previously. There are still several points that need to be considered before publication.

1. The updated and clarified treatment of the regulatory divergence categorization of the X-chromosome in hybrid males (described in lines 315-334, 939-963, and Figure S14) is now reasonable. However, I found some inconsistent statements regarding the trans-only category. In lines 326-328, the authors mention that they found only a non-significant trend in both sexes toward enrichment of trans-only regulatory divergence affecting X-linked genes, but in the Figure 3 legend, it says that the X-chromosome is enriched for trans-only changes in both sexes despite that Figure 3A does not show significant enrichment of trans-only category in either sex. Also, lines 695-698 (in Discussion) and line 746 (in Conclusion) refer to the trans-only effect being significant. In addition, there is no description about how "conserved" category indicated in Figure 3 was scored.

2. The numbering of the figures do not follow the order cited in the text. In line 285-293, Figure 5 is cited after Figure 2 and Figure 3 comes after Figure 4.

3. Lines 307-313 better fits in Discussion rather than in Results section.

Reviewer #2: Summary:

In this revision the authors have addressed major concerns regarding methods and interpretation with adjusted approaches and categorization. They have also done a wonderful job providing context, explanation of limitations and documentation that greatly improve the clarity and accessibility of the manuscript. I have only a couple comments, outlined below, and also provide minor suggestions for polish. The manuscript is much improved and close to being publication ready.

1. Pg. 13 lines 308-313. These are interesting mechanisms and should be included, but it seems like they would produce cis by trans effects or would not lead to dominance. Other possibilities might be transvection or differences in parent-of-origin effects?

2. Chromosome V has patterns that are similar to X, relative to other autosomes. Is there any other evidence that this chromosome could be derived from an ancestral X?

Minor suggestions

1. I suggest ‘cryptic evolution’ instead of ‘inconspicuous evolution’.

2. Pg 12 line 287, Figures are out of order – currently Fig 2, 5, 4, 3.

3. Pg 5 line 95, adding earlier references would be nice here (e.g. Yan et al. 2002).

4. Figure 4 and S10 – I think the labels or categories may have a typo. They don’t match the categories given in the methods.

5. Methods Pg. 34. It is a minor thing, but it might help to add that the modules are designated M1-M15 here.

Reviewer #3: The revised manuscript addresses most of the major reviewer concerns. The authors use gene expression data from C. briggsae her-1 mutants to control for the fact that briggsae hermaphrodites, but not nigoni or F1 females, have some spermatogenic cells. This approach and the way it informs the manuscript’s conclusions seems appropriate to me.

However, I am still not convinced that the assumptions required to infer modes of X-linked gene expression evolution in males are reasonable. The manuscript and response to reviews are clear about the assumptions being made, but does not adequately justify them, in my opinion. X-linked genes that are expressed in F1 males at levels similar to C. nigoni males are assumed to be the result of cis-regulatory evolution, but this will include (an unknown proportion of) genes regulated by C. nigoni trans-factors that are dominant in the F1. Similarly, the inferences about cis-trans compensatory evolution are confounded by under- and over-dominance just among trans-factors. The previous studies that are referenced to support their justifications don’t seem to me to be appropriate, as Witkopp 2004 and McManus 2012 used only hybrid females, and Wayne 2004 used a balanced round-robin crossing design, so at a minimum I think there needs to be a better explanation of how those studies justify the authors’ current assumptions. In short, without two alleles from which to measure ASE, or at least reciprocal males carrying different X chromosomes in the same F1 autosomal genotype, I’m not confident that much can be said about whether gene expression changes are the result of cis- or trans-divergence. I think these results, and the attendant conclusions regarding the large X-effect and hybrid male sterility, need to be revised or possibly removed.

line 138: is inclusion really the best word here?

line 179: “consistent” here seems to indicate that the pattern is the same across the different autosomes, but in the next half of the sentence it seems to indicate that the proportion of genes with higher expression of the nigoni allele vs. higher expression of the the briggsae allele are equal. If the intention is the latter, I suggest choosing a different word to make this clearer.

lines 234-236: some numbers are randomly underlined

line 347: I don’t see the connection between the observation that genes with transgressive expression in F1’s are more likely to show compensatory cis-trans interactions and “stabilizing selection can act differently on sex-specific transcript abundance”. This conclusion should be more explicitly explained. The following sentence (line 349) makes it sound like the previous sentences were reporting results for genes with sex-biased expression (?) but that is not stated above.

line 380: How do the authors distinguish between the hypotheses that 1) for the same number of mutational substitutions, the downstream effects on the expression of male-biased genes are greater than the effects on female-biased genes (this is my interpretation of “fragile”), vs. 2) the factors controlling the expression of male-biased genes evolve more rapidly than those controlling female-biased genes? Perhaps it would help to elaborate on what exactly they mean by “fragile”. The same question applies to the term “resilience” in the following sentences.

line 418: The hypothesis that genes involved in spermatogenesis are depleted from the Drosophila X chromosome has been refuted (Meiklejohn & Presgraves 2012 GBE 4:895-904; Meisel et al 2012 PLoS Gen 8(10):e1003013 ) or is at least contentious.

lines 428-431: Are these results only considering gene expression in hermaphrodites? This is implied in Figure 6 where it appears panels D-G only include data for hermaphrodites; and makes sense considering that ASE is not possible in males for X-linked genes.

line 527: I don’t see how the authors draw conclusions about the “sexually-dimorphic transcriptomes” from Figure 4, which does not break genes out by their expression differences between males and females. In general, the language throughout the manuscript should maybe be tightened and formalized a bit, to help the reader discriminate between genes differentially expressed between the sexes (that is what I think of when I read sexually-dimorphic transcriptomes) and other patterns that differ between males and females, such as gene inheritance or evolution. This happens again on line 583 which suggests the data in Figure 4 is somehow broken out by male- or female-biased expression, which I do not see.

In the equation on line 908, one of the “Cbr" should be a “Cni” I think

**Have all data underlying the figures and results presented in the manuscript been provided?**

Reviewer #1: Yes

Reviewer #2: Yes

Reviewer #3: Yes

PLOS authors have the option to publish the peer review history of their article (what does this mean?). If published, this will include your full peer review and any attached files.

Reviewer #1: No

Reviewer #2: No

Reviewer #3: No

---

## [Editor Report · Decision Letter 1]

9 Feb 2021

Dear Dr Sanchez-Ramirez,

We are pleased to inform you that your manuscript entitled "Widespread misregulation of inter-species hybrid transcriptomes due to sex-specific and sex-chromosome regulatory evolution" has been editorially accepted for publication in PLOS Genetics. Congratulations!

Yours sincerely,

Alex Buerkle

Associate Editor

PLOS Genetics

Bret Payseur

Section Editor: Evolution

PLOS Genetics

Comments from the reviewers (if applicable):

I appreciate the authors' further attention to suggestions for improvement of the manuscript, including refining wording in the manuscript, correcting small errors, and clearly laying out their responses to the points raised by reviewers.

**Data Deposition**

http://datadryad.org/submit?journalID=pgenetics&manu=PGENETICS-D-20-01810R1

**Press Queries**

---

## [Editor Report · Acceptance letter]

1 Mar 2021

PGENETICS-D-20-01810R1 

Widespread misregulation of inter-species hybrid transcriptomes due to sex-specific and sex-chromosome regulatory evolution 

Dear Dr Sanchez-Ramirez, 

We are pleased to inform you that your manuscript entitled "Widespread misregulation of inter-species hybrid transcriptomes due to sex-specific and sex-chromosome regulatory evolution" has been formally accepted for publication in PLOS Genetics! Your manuscript is now with our production department and you will be notified of the publication date in due course.

With kind regards,

Alice Ellingham

PLOS Genetics

On behalf of:
